# Mitochondrial phosphatase PGAM5 modulates cellular senescence by regulating mitochondrial dynamics

Bo Yu[1], Jing Ma[1], Jing Li[2], Dazhi Wang[3], Zhigao Wang[4] & Shusheng Wang [1,5✉]

Mitochondria undergo dynamic fusion/fission, biogenesis and mitophagy in response to stimuli or stresses. Disruption of mitochondrial homeostasis could lead to cell senescence, although the underlying mechanism remains unclear. We show that deletion of mitochondrial phosphatase PGAM5 leads to accelerated retinal pigment epithelial (RPE) senescence in vitro and in vivo. Mechanistically, PGAM5 is required for mitochondrial fission through dephosphorylating DRP1. PGAM5 deletion leads to increased mitochondrial fusion and decreased mitochondrial turnover. As results, cellular ATP and reactive oxygen species (ROS) levels are elevated, mTOR and IRF/IFN-β signaling pathways are enhanced, leading to cellular senescence. Overexpression of Drp1 K38A or S637A mutant phenocopies or rescues mTOR activation and senescence in PGAM5$^{-/-}$ cells, respectively. Young but not aging Pgam5$^{-/-}$ mice are resistant to sodium iodate-induced RPE cell death. Our studies establish a link between defective mitochondrial fission, cellular senescence and age-dependent oxidative stress response, which have implications in age-related diseases.

[1] Department of Cell and Molecular Biology, Tulane University, New Orleans, LA 70118, USA. [2] Department of Bioengineering, Rice University, Houston, TX 77030, USA. [3] Department of Cardiology, Boston Children's Hospital, Harvard Medical School, Boston, MA 02115, USA. [4] Department of Molecular Biology, UT Southwestern, Dallas, TX 75390, USA. [5] Department of Ophthalmology, Tulane University, New Orleans, LA 70118, USA. ✉email: swang1@tulane.edu

Cellular senescence is a phenomenon originally defined as stable cell cycle arrest in response to different stresses[1]. Besides cell cycle exit, senescent cells undergo phenotypic changes including metabolic reprogramming, chromatin rearrangement and autophagy modulation[2]. Features of senescence include increased cellular volume, shortened telomeres, increased reactive oxygen species (ROS) level, persistent DNA damage response, elevated senescence-associated β-galactosidase activity, p16[Ink4A] expression, etc. Senescent cells produce and secrete a combination of factors to exert non-cell-autonomous effects, referred to as senescence-associated secretory phenotype (SASP). Senescence can be a protective mechanism against stress or cancer. However, accumulation of senescence cells could drive aging and age-related diseases. Age-related macular degeneration (AMD) is a degenerative disease of the retina and the leading cause of irreversible blindness in the elderly[3]. Dry AMD is characterized by degeneration of retinal pigment epithelial (RPE) layer at late stage. The role of RPE senescence in AMD remains unclear[4].

Mitochondria has critical roles in cellular senescence due to ROS generation from oxidative phosphorylation[5]. It undergoes dynamic fusion/fission, biogenesis and mitophagy in response to physiological stimuli or pathological stresses. The role of mitochondrial fusion/fission in aging (or senescence) has been controversial. Inhibition of mitochondrial fission by deleting a mitochondrial fission protein dynamin-related protein 1 (Drp1) or maintenance of the fused mitochondrial network is necessary for longevity in yeast or *C. elegans*[6,7]. However, disruption of either mitochondrial fission or fusion significantly reduces medium lifespan in *C. elegans*, while promoting Drp1-mediated mitochondrial fission in midlife prolongs healthy lifespan of *Drosophila melanogaster*[8,9]. These controversies suggest the necessity to clarify mitochondrial fission/fusion function in aging (senescence) and longevity, especially in higher vertebrate species.

Mitchondrial fission/fusion, mitophagy and biogenesis collectively control mitochondrial turnover. Mitophagy, a process of selective engulfment of dysfunctional mitochondria for degradation by lysosome, is indispensable for maintaining cellular homeostasis. Impaired mitophagy indicates less mitochondrial turnover, which leads to accumulation of dysfunctional mitochondria, senescence and age-related disorders[10]. Mitophagy is orchestrated by mitochondrial dynamics[11]. Drp1-driven mitochondrial fission facilitates mitophagy by dividing mitochondria into fragments or segregating damaged mitochondrial subdomains for autophagosome engulfment[12,13].

Phosphoglycerate mutase 5 (PGAM5) is a mitochondrial Serine (Ser)/Threonine (Thr) phosphatase normally located in the inner mitochondrial membrane. Upon mitochondrial dysfunction, PGAM5 recruits and dephosphorylates Drp1 at Ser-637, triggers its GTPase activity and promotes mitochondrial fission[14,15]. PGAM5 regulates mitophagy by stabilizing PINK1 under stress conditions, which recruits E3 ubiquitin ligase PARKIN for degradation of the damaged mitochondria[15,16]. PGAM5 also regulates mitophagy independent of PARKIN by interacting and dephosphorylating FUNDC1, which interacts with LC3 (ref. [17]). In the end, PGAM5 can be cleaved and released to the cytoplasm through PARKIN, which activates Wnt signaling and induces mitochondrial biogenesis[18]. PGAM5 also regulates anti-oxidative response by forming a tertiary complex with KEAP1 and NRF2 (ref. [19]). It regulates necroptosis by acting as a RIPK3 target and recruiting the RIPK1-RIPK3-MLKL necrosis "attack" complex to mitochondria[14]. *Pgam5*[−/−] mice show Parkinson-like movement disorder[16]. In sum, PGAM5 has multiple functions and could act as signaling hub to sense mitochondrial stress, regulate mitochondrial dynamics and anti-oxidative response.

Given the importance of PGAM5 in mitochondrial dynamics, we ask whether PGAM5 regulates cellular senescence and age-dependent anti-oxidative response. Through in vitro and in vivo approaches, we show that PGAM5 is essential for mitochondrial homeostasis, and *Pgam5* deficiency induces accelerated senescence in mice. PGAM5 deletion leads to reduced mitochondrial turnover, increased ATP and ROS levels, elevated mTOR and IRF/IFN-β signaling pathways. Collectively, these result in cellular senescence and age-related reduction in anti-oxidation capability. ATP elevation, mTOR activation and senescence in *PGAM5*[−/−] cells can be phenocopied or rescued by overexpression of DRP1-K38A or S637A mutant, respectively, underscoring the significance of mitochondrial fission in repressing cellular senescence.

## Results

**PGAM5 deletion leads to accelerated cell senescence.** To determine the role of mitochondrial fusion/fission in mouse aging, *Pgam5 flox/flox* mice were generated using verified ES cell from European Mouse Mutant Cell Repository, and *Pgam5*[−/−] allele was made by crossing *Pgam5 flox/flox* mice (C57BL/6J background) with *CAG-Cre* mice (Fig. 1a). As a Laz cassette was infused between exon I and exon II in *Pgam5 flox* or knockout allele, Laz staining was used to monitor PGAM5 protein expression. In the retina, LacZ signal was enriched in the retinal pigment epithelium (RPE) layer, retinal ganglion cells and ciliary body epithelium (Fig. 1b). Successful knockout (KO) of *Pgam5* in the mice was confirmed by western blot analysis using PGAM5 antibody (Fig. 1c). *Pgam5*[−/−] mice were born at Mendel ratio, but weighed significantly less compared to the wild-type (WT) controls at 1-year and older (Supplementary Fig. 1a, b). ~50% of the *Pgam5*[−/−] mice (aged 1–1.5 years) showed lordokyphosis (hunchback), odd gait and swollen foot, and among them ~50% died from unknown causes or were too sick and killed as required by animal facility (see Kaplan–Meier survival curve in Fig. 1d). These phenomena were not observed in age-matched WT control mice. Aged *Pgam5*[−/−] mice appeared to have fewer movements under a new environment compared with control mice (Supplementary Video 1), consistent with the Parkinson's-like movement disorders observed previously[16]. Together, these suggest an accelerated senescence (or aging) phenotype in *Pgam5*[−/−] mice.

To confirm the senescence-related phenotype in *Pgam5*[−/−] mice, features of cellular senescence were examined. Cell size increases during cell senescence[20]. RPE cell size in 2- and 18-month-old mice was quantified after ZO-1 staining. Although they are similar in 2-month-old mice, it was more increased in 18-month-old *Pgam5*[−/−] mice compared to the controls (Fig. 1e). Phospho-rH2AX is upregulated in response to DNA damage and accumulated in senescent cells[21]. Compared to the controls, significantly higher phospho-rH2AX-positive RPE cell ratio was observed in 18-month-old *Pgam5*[−/−] mice. Of note, phospho-H2AX positive cells also showed karyorrhexis (destructive fragmentation of nucleus) in both wild-type and KO mice, suggesting cellular senescence (Fig. 1f, g). Similar results were also observed in the skin (Fig. 1j, middle panels). macroH2A and P16[Ink4a] are hallmarks for cellular senescence. macroH2A was detected in the RPE cells of 18-month-old WT mice, which was significantly increased in the age-matched *Pgam5*[−/−] mice (Fig. 1h, i). Strong P16[Ink4a] expression were observed in the retina and the skin of 18-month-old *Pgam5*[−/−] mice, but not in the WT controls (Fig. 1h, j right panel). Senescence cells also show increased p53 but decreased Lamin B1 expression, as well as senescence-associated secretory phenotype (SASP) as shown by elevated *Il-6*, *Mmp3* and *Tnfα* expression[22–24]. Indeed, increased MMP3, p53 and decreased Lamin B1 protein expression was

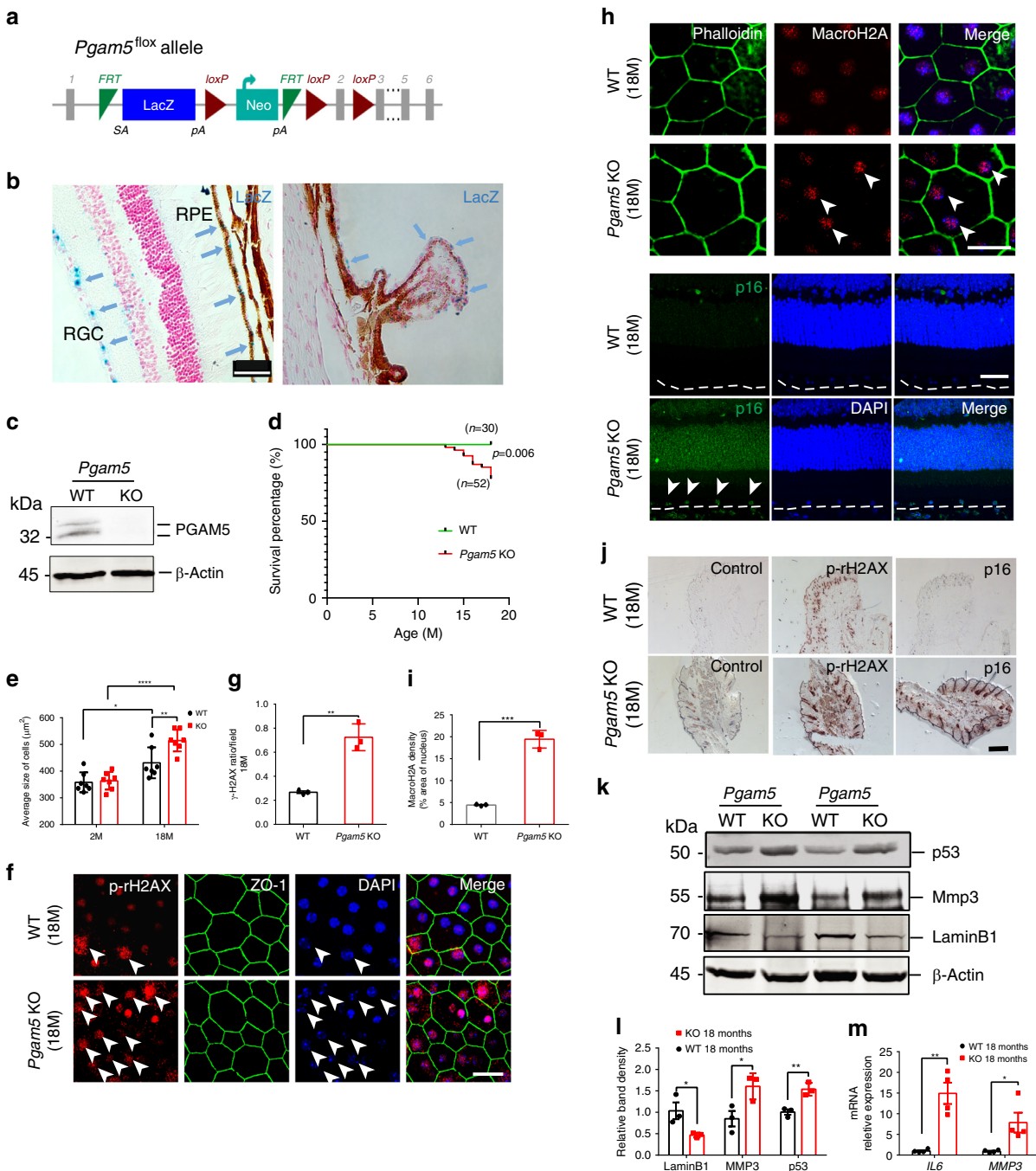

**Fig. 1 Cellular senescence induced by *PGAM5* deletion in vivo. a** Schematic of the *Pgam5* flox allele with LacZ location noted. Exon 2 is flanked by loxP; **b** β-gal (lacZ) activity detected in *Pgam5*−/− mice. High LacZ activities were shown in RPE, RGC and ciliary body epithelium. Scale bar: 100 μm. **c** Western blot confirming PGAM5 deletion in vivo (β-Actin as loading control). n = 3. **d** Kaplan–Meier survival analysis of *Pgam5*−/− mice. Animal numbers were shown, p = 0.006, Log-rank (Mantel–Cox) test. **e** RPE cell size from WT and *Pgam5*−/− mice at 2 and 18 months. n = 7. *p = 0.0263, **p < 0.005, ****p < 0.0001, two-way ANOVA, Tukey's multiple comparisons test. Error bars, mean ± s.d. **f** Flatmount phopho-γH2AX staining in the RPE layer of WT and *Pgam5*−/− mice at 18 months n = 3. Scale bar: 20 μm. Arrows point to the nucleus with karyorrhexis. **g** Statistics of phospho-γH2AX ratio/each field in WT and *Pgam5*−/− mice at 18 months. n = 3. **p = 0.0022, two-tailed unpaired t-tests. Error bars, mean ± s.d. **h** MacroH2A, Phalloidin and P16[Ink4a] immunostaining in WT and *Pgam5*−/− Retina/RPE sections at 18-months-old. Scale bar: 20 μm (macroH2A) or 50 μm (p16). n = 3. **i** Quantified macroH2A density in WT and *Pgam5*−/− RPE cells. n = 3. ***p = 0.0002, two-tailed unpaired t-tests. Error bars, mean ± s.d. **j** Immunohistochemistry of phospho-γH2AX and P16[Ink4a] in WT and *Pgam5*−/− mice skin at 18-months-old. In control group, no primary antibodies were used. Scale bar: 500 μm. n = 3. **k** Western blots of p53, Lamin B1 and MMP3 from RPE/choroid sample in WT and *Pgam5*−/− mice (β-Actin as loading control). n = 3; **l** Quantification of p53, Lamin B1 and MMP3 bands relative to β-Actin in **k**. n = 3, *p = 0.0244 for Lamin B1, p = 0.04 for MMP3, **p = 0.0077 for p53, two-tailed unpaired t-tests. Error bars, mean ± s.d. **m** Il-6 and Mmp3 mRNA level in the RPE/choroid of WT and *Pgam5*−/− mice at 18 months. n = 4, **p = 0.0016, *p = 0.0276, two-tailed unpaired t-tests. Error bars, mean ± s.e.m. For assays in the figure, n represents the number of biologically independent experiments. Images were captured under same settings, and representative images were shown. Source data are available as a Source Data file.

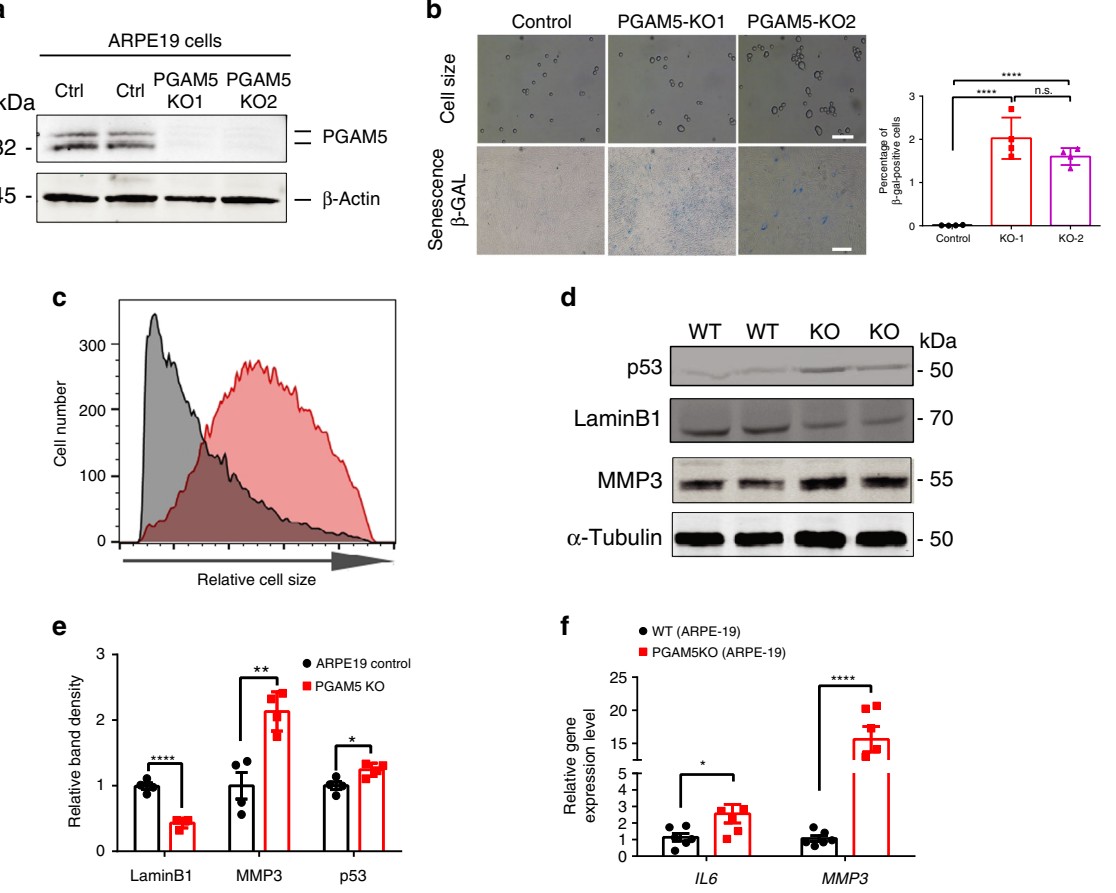

**Fig. 2 Cellular senescence induced by *PGAM5* deletion in vitro. a** Western blot confirming PGAM5 deletion in ARPE-19 cells using CRISPR/Cas9 technology. β-Actin was used as a loading control. $n = 3$. **b** Top panel: cell size change detected in $PGAM5^{-/-}$ ARPE-19 cells after 8 weeks culture and trypsinization; bottom panel: SA-β-gal activity change detected in $PGAM5^{-/-}$ ARPE-19 cells after 8 weeks culture; quantification of β-gal staining was shown in the bar graph (right panel). $n = 4$. Scale bar = 100 μm (up panels) or 500 μm (down panels). ****$p < 0.0001$, one-way ANOVA Tukey's multiple comparisons test. Error bars, mean ± s.d. **c** Relative cell size distribution in WT and $PGAM5^{-/-}$ ARPE-19 cells after 8 weeks culture as measured by flow cytometry. $X$ axis is FSC-A, which reflects cell size. $n = 3$. **d** Western blots confirming p53 and MMP3 upregulation and Lamin B1 downregulation $PGAM5^{-/-}$ AREP-19 cells after 8 weeks culture. α-Tubulin was used as a loading control $n = 4$. **e** Quantification for the bands density in **d**. $n = 4$, ****$p < 0.0001$, **$p = 0.0041$, *$p = 0.0207$, two-tailed unpaired $t$-tests; error bars, mean ± s.d.; **f** *IL6, MMP3* mRNA level as measured by qRT-PCR in WT and $PGAM5^{-/-}$ ARPE-19 cells after 8 weeks culture. $n = 6$, *$p < 0.05$, ****$p < 0.0001$, two-tailed unpaired $t$-tests. Error bars, mean ± s.e.m. For assays in the figure, $n$ represents the number of biologically independent experiments. Images were captured under same settings, and representative images were shown. Source data are available as a Source Data file.

observed in the RPE/choroid in 18-month-old $Pgam5^{-/-}$ mice by western blot analysis (Fig. 1k, l). *Mmp3* and *Il-6* RNA levels were upregulated by ~5 and 15 folds, respectively, in the 18-month-old RPE/choroid of $Pgam5^{-/-}$ mice compared with age-matched controls. (Fig. 1m). *Tnfα* expression was too low to detect in those samples. Taken together, these indicate an accelerated senescent phenotype in $Pgam5^{-/-}$ mice.

Two independent $PGAM5^{-/-}$ ARPE-19 cell lines were established by using CRISPR-Cas9 technology to confirm their senescent phenotypes in vitro. Efficient $PGAM5^{-/-}$ cells was confirmed by western blot analysis (Fig. 2a). When $PGAM5^{-/-}$ ARPE-19 cells were cultured for 8 weeks, elevated β-galactosidase (β-gal) activity was observed compared to the controls (Fig. 2b). Moreover, cell size distribution indicates that $PGAM5^{-/-}$ cells have larger cell volume (Fig. 2b, c, Supplementary Fig. 1c). Similar to in vivo, upregulated MMP3, p53 and downregulated Lamin B1 protein expression was observed in $PGAM5^{-/-}$ ARPE-19 cells compared to controls after 8 weeks culture (Fig. 2d, e). RNA level of SASP markers *IL6* and *MMP3* was increased by ~2.5 and 15 folds in the $PGAM5^{-/-}$ cells, respectively (Fig. 2f). TNFα was

undetectable in these cells. Together, these confirmed accelerated senescent phenotype in $PGAM5^{-/-}$ ARPE-19 cells.

**PGAM5 deletion induces changes in mitochondrial dynamics.** To explore the underlying mechanism of *PGAM5* deletion-induced cellular senescence, mitochondrial morphology and dynamics were initially evaluated[14]. Compared to the controls, mitochondria of $PGAM5^{-/-}$ ARPE-19 cells exhibited more tubular and less fragmented structures, as quantified by a ~30% increase in the average length of Tom20-positive mitochondrial branches (Fig. 3a and Supplementary Fig. 2a). Consistent with PGAM5 regulation of mitochondrial fission by dephosphorylating Drp1, Drp-1 (Ser-637) phosphorylation was increased in $PGAM5^{-/-}$ ARPE-19 cells, supporting that PGAM5 is required for mitochondrial fission via dephosphorylating Drp1 (Fig. 3b, and Supplementary Fig. 2b).

PGAM5 has been reported to have long and short forms, as well as full-length and cleaved forms[14]. Cleaved PGAM5 retains its phosphatase domain[25], and could release from mitochondria

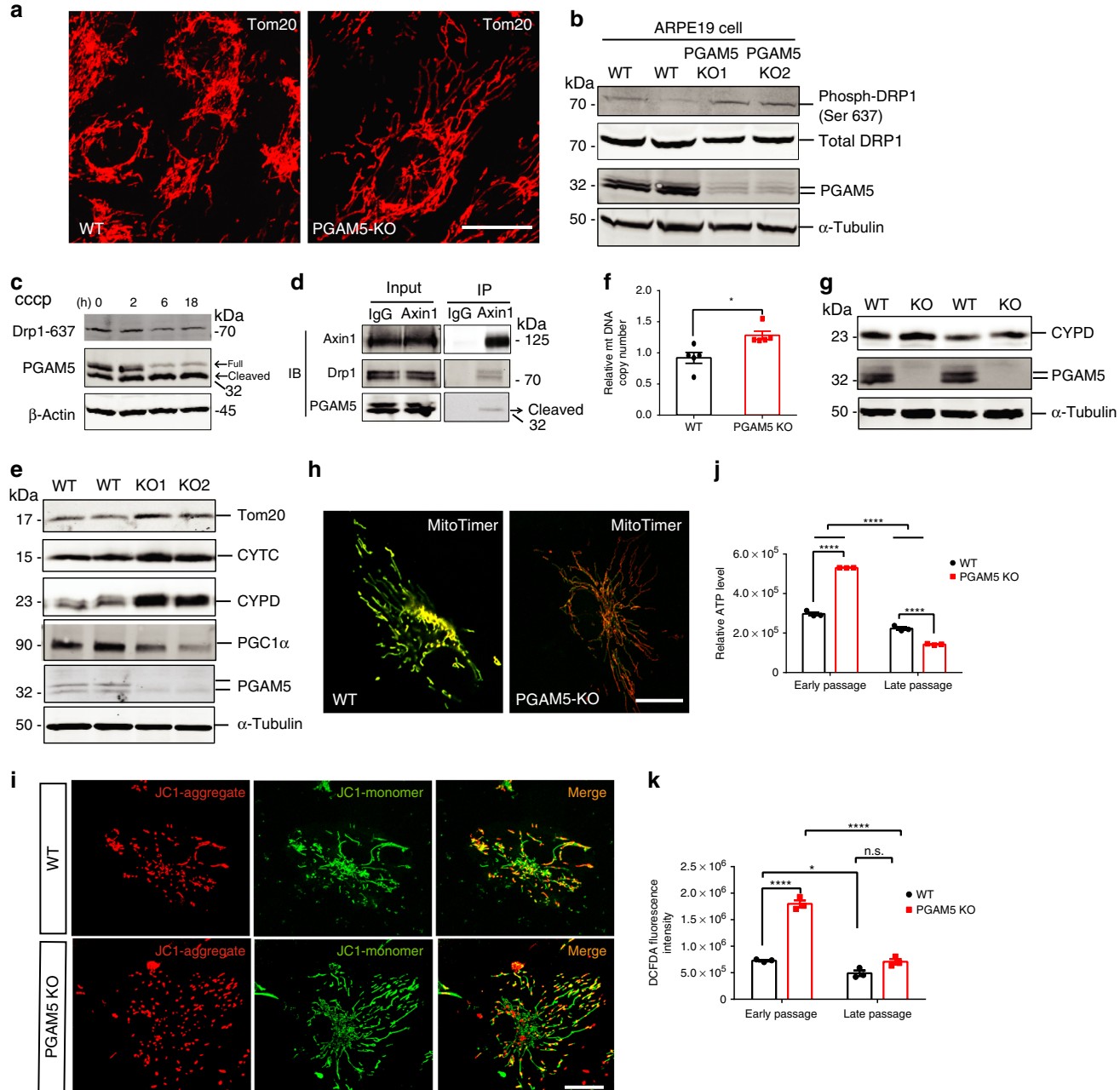

**Fig. 3 Mitochondrial dynamics and function change upon *PGAM5* deletion. a** Mitochondrial morphology outlined by Tom20 antibodies in control and *PGAM5*−/− ARPE-19 cells. Scale bar = 20 μm. *n* = 5. **b** Western blots showing upregulation of phosphor-Drp1(S637) but not total Drp1 in *PGAM5*−/− ARPE-19 cells. α-Tubulin was used as loading control. *n* = 4. **c** Western blots showing PGAM5 cleavage and Drp1(S637) dephosphorylation in ARPE-19 cells by CCCP treatment. *n* = 3. **d** Co-immunoprecipitation experiment using Axin1 antibody, showing that Axin1 interacts with both Drp1 and cleaved PGAM5 in ARPE-19 cells. *n* = 3. **e** Western blots showing upregulation of mitochondrial proteins (Tom20, CYTC, CYPD) and downregulation of PGC1α in *PGAM5*−/− ARPE-19 cells. α-Tubulin was used as loading control. *n* = 4. **f** Increased mitochondrial DNA in *PGAM5*−/− ARPE-19 cells. *n* = 5. *$p = 0.0111$, two-tailed unpaired *t*-tests; error bars, mean ± s.e.m. **g** Increased mitochondrial protein Cypd in the RPE/choroid of *Pgam5*−/− mice. α-Tubulin was used as loading control. *n* = 3. **h** Decreased mitochondrial turnover in *PGAM5*−/− ARPE-19 by MitoTimer transfection and labeling. Scale Bar equals to 20 μm. *n* = 3. **i** Mitochondrial membrane potential change as labeled by JC-1 in WT and *PGAM5*−/− ARPE-19 cells. Scale bar = 20 μm. *n* = 5. **j** ATP level change as measured in short-term (1 week) and long-term (8 weeks) culture of WT and *PGAM5*−/− ARPE-19 cells. *n* = 3, ****$p < 0.0001$, two-way ANOVA Tukey's multiple comparisons test; error bars, mean ± s.e.m. **k** ROS change as measured in short-term (1 week) and long-term (8 weeks) culture of WT and *PGAM5*−/− ARPE-19 cells. *n* = 3, *$p < 0.05$; ****$p < 0.0001$, two-way ANOVA Tukey's multiple comparisons test. n.s. represents no significance. Error bars, mean ± s.e.m.; for assays in the figure, *n* represents the number of biologically independent experiments. Images were captured under same settings, and representative images were shown. Source data are available as a Source Data file.

to cytosol to interact with Axin1 (ref. [18]). Moreover, dephosphorylation of Drp1-Ser-637 is essential for its binding to mitochondria[26]. Based on these, we tested the hypothesis that cleaved PGAM5 interacts with Axin1 and recruits Drp1 for dephosphorylation and fission processing. Both full-length and cleaved PGAM5 forms were found is ARPE-19 cells, with cleaved form dominant when induced by mitochondrial uncoupling agent carbonyl cyanide chlorophenylhydrazone (CCCP), consistent a decrease in Drp1 (Ser-637) phosphorylation (Fig. 3c). A shorter cleaved form the claimed "short form" was not detected, disapproving the existence of the PGAM5 short form. By co-immunoprecipitation assay using antibody to Axin1, both Drp1 and cleaved PGAM5 can be co-immunoprecipitated with Axin1 (Fig. 3d). Co-localization of Axin1, Drp1 and PGAM5 was also confirmed by immunofluorescence, supporting that cleaved PGAM5 recruits and dephosphorylates Drp1 through interacting with Axin1 (Supplementary Fig. 2c).

As Drp1-mediated mitochondrial fission is required for mitochondrial homeostasis, we hypothesized that hyperfusion from PGAM5 deficiency could result in less mitochondrial turnover[11,15,27–29]. The mitochondrial outer membrane protein Tom20, inner membrane-associated protein cytochrome $C$, and mitochondrial matrix protein, Cyclophilin F (CYPD), were all increased by 20–40% in $PGAM5^{-/-}$ ARPE-19 cells, indicating more mitochondrial mass (Fig. 3e, and Supplementary Fig. 2d). Consistently, the mitochondrial DNA copy number was also increased in $PGAM5^{-/-}$ ARPE-19 cells (Fig. 3f). The increase in mitochondrial mass was also confirmed in vivo by western blot using CYPD antibody in RPE/choroid tissue isolated from $Pgam5^{-/-}$ mice (Fig. 3g). To test whether mitochondrial biogenesis contributes to increased mitochondrial mass in $PGAM5^{-/-}$ RPE cells, the protein level of PGC1α, a critical cotranscriptional regulator for mitochondrial biogenesis, was examined. PGC1α was significantly downregulated by $PGAM5$ deletion, arguing against the possibility that increased mitochondrial biogenesis contributes to increased mitochondrial mass in $PGAM5^{-/-}$ cells (Fig. 3e). Instead, more mitochondrial mass and less mitochondrial biogenesis support less mitochondria turnover by $PGAM5$ deletion. To directly monitor mitochondrial turnover, MitoTimer labeling was used[30]. MitoTimer is a mitochondria-targeting florescence protein that is green once being translated and becomes more red over time. As expected, $PGAM5$ deletion induced more "Red" mitochondria accumulation than control cells, supporting less mitochondrial turnover (Fig. 3h). Mitochondrial membrane potential ($\Delta\Psi_M$) is an important parameter for mitochondrial function. JC-1 dye was used to measure the $\Delta\Psi_M$ change, JC-1 could distinguish between low and high $\Delta\Psi_M$ in mitochondria. The high membrane potential allows JC-1 to form red J-aggregates, but in low membrane potential mitochondria, JC-1 forms green J-monomers. Increased number of unhealthy "green" labeled low membrane potential mitochondria were observed in $PGAM5^{-/-}$ ARPE-19 cells compared to the control cells (Fig. 3i and Supplementary Fig. 2e). Moreover, the proportion of the "red" high membrane potential mitochondria was increased in $PGAM5^{-/-}$ ARPE-19 cells even after normalization to the "green" signal (Fig. 3i and Supplementary Fig. 2f). These indicate that both "healthy" (red) and "unhealthy" (green) mitochondria are increased in $PGAM5^{-/-}$ ARPE-19 cells due to the accumulation in total mitochondrial mass. Together, our data suggested $PGAM5$ deletion leads to mitochondrial hyperfusion and less mitochondrial turnover.

To examine the functional consequence of the mitochondrial morphological and dynamic changes, mitochondrial ATP and reactive oxygen species (ROS) production was measured. Consistent with the increased mitochondrial mass and less turnover, $PGAM5^{-/-}$ ARPE-19 cells under short-term culture (1 week) produced 67% more ATP compared to the controls (Fig. 3j). However, the increased production of ATP did not sustain overtime. $PGAM5^{-/-}$ RPE under long-term (8 weeks) culture produced ~30% less ATP compared to the controls. Elevated ROS could contribute to cellular senescence[5]. Cellular ROS level was increased ~50% in $PGAM5^{-/-}$ ARPE-19 cells under short-term culture, but significantly decreased in cells under long-term culture as shown by DCFDA dye staining (Fig. 3k).

**Regulation of AMPK–mTOR pathway by PGAM5.** As ATP was elevated in $PGAM5^{-/-}$ ARPE-19 cells, AMPK–mTOR signaling pathway was examined as underlying mechanism for PGAM5-regulated senescence. Cellular AMP:ATP ratio reflects the energy requirement of a cell, and can be sensed by intracellular sensor AMP-activated protein kinase (AMPK)[31,32]. AMPK phosphorylation but not total AMPK level was reduced by more than 50% in $PGAM5^{-/-}$ ARPE-19 cells cultured for a week (Fig. 4a). TSC2, a negative modulator of mTORC1, is regulated by AMPK[33,34]. As expected, TSC2 phosphorylation was significantly reduced, and mTORC1 activity indicated by the phosphorylation of S6 ribosomal protein was markedly boosted in those cells. These were confirmed by PGAM5 siRNA experiments. mTOR pathway can be activated by protein biosynthesis inhibitor cycloheximide (CHX) treatment in MEF and HEK293 cells[35]. We found CHX stimulates mTORC1 pathway in ARPE-19 cells. Higher basal level and stronger CHX-induced S6 and 4EBP1 phosphorylation were observed upon PGAM5 silencing (Fig. 4b). mTOR pathway activation by PGAM5 silencing was not restricted to ARPE-19 cells and was also observed in primary human RPE (HRPE) and human umbilical vein endothelial cells (HUVEC) (Supplementary Fig. 3a, b). Upregulation of S6 phosphorylation was also confirmed in vivo by western blot of RPE/choroid tissue and immunostaining of retinal tissue in 2-month-old $Pgam5^{-/-}$ mice (Fig. 4c, d). To further validate the causative relationship between ATP elevation and mTORC1 activation, ATPase inhibitor oligomycin A or ROS inhibitor N-acetyl-L-cysteine (NAC) was used to treat $PGAM5^{-/-}$ RPE cells. Activation of mTOR signaling was blunted by oligomycin A but not NAC (Fig. 4e). This supports that the elevated ATP level but not ROS is responsible for mTOR activation in $PGAM5^{-/-}$ RPE cells. These data collectively indicate that ATP elevation by $PGAM5$ deletion stimulates mTORC1 activity, which could explain the accelerated senescence in $PGAM5^{-/-}$ cells[36].

**Drp1-K38A overexpression mimicks $PGAM5^{-/-}$ cell phenotypes.** We asked further whether the increased Drp1 phosphorylation drives mitochondrial morphological changes and mTOR activation in $PGAM5^{-/-}$ cells. As Drp1 Ser-637 dephosphorylation is required for its GTPase activity, higher Drp1 phosphorylation indicates less GTPase activity. We reasoned that Drp1-K38A mutant, which loses GTPase activity, could theoretically phenocopy $PGAM5^{-/-}$ phenotypes[26,37]. To test this, Drp1-K38A mutant was overexpressed in RPE cells using adenoviruses. As expected, Drp1-K38A overexpression attenuated mitochondrial fission and elongated mitochondrial branches by almost 2-fold in WT cells, similar to what was observed in $PGAM5^{-/-}$ ARPE-19 cells (Fig. 5a, b). This indicates that Drp1-K38A overexpression mimics $PGAM5^{-/-}$mitochondrial hyperfusion phenotype. Consistently, cellular ATP level was significantly increased in both WT and $PGAM5^{-/-}$ cells by Drp1-K38A overexpression (Fig. 5c, d). To further confirm the effect of Drp1-K38A overexpression on RPE senescence, HRPE (primary human RPE cell) cells were infected with Ad-GFP or Ad-Drp1-K38A (MOI of 100). mTORC1 activation in Drp1-K38A overexpressed HRPE cells was

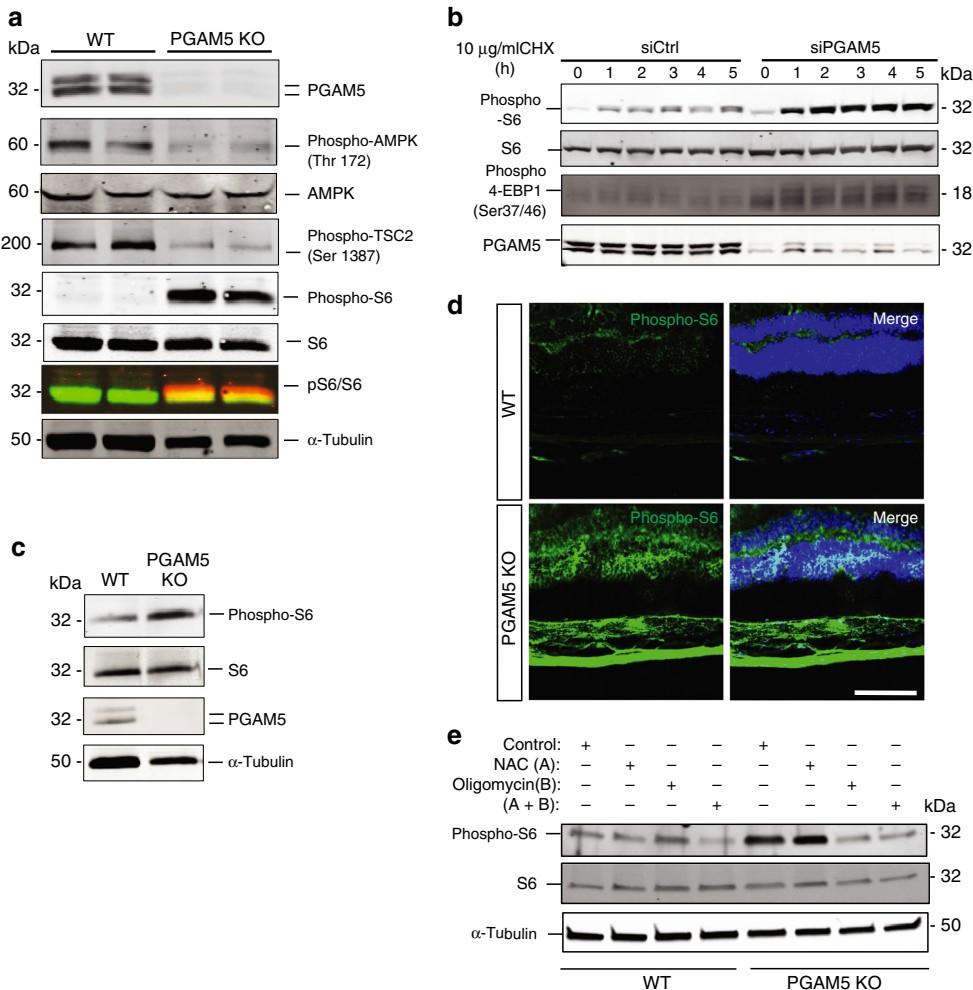

**Fig. 4 Regulation of AMPK–mTOR pathway by PGAM5. a** Expression of indicated proteins (PGAM5, phosphor-AMPK, total AMPK, phosphor-TSC2, phosphor-S6), total S6 in WT and *PGAM5*$^{-/-}$ ARPE-19 cells by western blot after medium change for 48 h. Cells were cultured for 1 week. α-Tubulin was used as loading control. $n = 3$. **b** Expression of indicated proteins (PGAM5, phosphor-S6, total S6 and phosphor-4EBP1) in WT and *PGAM5 silenced* ARPE-19 cells by western blot after treatment with 10 μg/ml CHX for indicated times. $n = 3$. **c** Representative western blots showing increased phosphor-S6 but not total S6 protein in the RPE/choroid of *Pgam5*$^{-/-}$ mice. α-Tubulin was used as loading control. $n = 3$. **d** Increased phosphor-S6 in retinal cryosections of *Pgam5*$^{-/-}$ mice (2-months-old) compared to age-matched WT mice. Scale bar = 100 μm. $n = 3$. **e** Phosphor-S6 and total S6 protein expression in WT and *PGAM5*$^{-/-}$ ARPE-19 cells by western blot after medium change for 48 h and NAC (3 mM) and/or oligomycin (5 μM) treatment for overnight. α-Tubulin was used as loading control. $n = 3$. For assays in the figure, *n* represents the number of biologically independent experiments. Images were captured under same settings, and representative images were shown. Source data are available as a Source Data file.

confirmed by western blot with phospho-S6 antibody (Fig. 5e and Supplementary Fig. 4a). Significantly stronger SA-β-Gal staining was observed in Drp1-K38A overexpressed cells compared to the controls at 5 weeks after infection (Fig. 5f). When the Drp1-K38A overexpressed cells were culturing for 8 weeks, increased p16 $^{INK4A}$ and MMP3 but decreased Lamin B1 protein expression was observed, supporting accelerated cell senescence (Fig. 5g, h). As to SASP markers, *IL6* RNA level was upregulated ~6 folds, and MMP3 mRNA level was elevated for more than 50 folds in the Drp1-K38A group. Of note, TNFα, which was undetectable in ARPE-19 cells and mouse RPE/choroid, was also increased by ~20 folds in HRPE cells, consistent with the heterogeneity of SASP markers in different systems (Fig. 5i). These data indicate that Drp1-K38A overexpression phenocopies the mitochondrial morphological change, mTOR activation and senescence phenotype driven by PGAM5 deficiency.

**Drp1-S637A overexpression rescues *PGAM5*$^{-/-}$ phenotypes.** We next examined whether overexpression of Drp1 S637A

phosphorylation mutant rescues the accelerated senescent phenotypes in *PGAM5*$^{-/-}$ cells[38]. Lentivirus expressing Drp1 S637A was used to transduce ARPE-19 cell at MOI of 10. Drp1 S637A overexpression markedly inhibited mitochondria fusion in *PGAM5*$^{-/-}$ ARPE-19 cells but not WT controls (Fig. 6a). By quantification, mitochondrial branch length was reduced to ~50% in *PGAM5*$^{-/-}$ ARPE-19 cells but not significantly changed in WT cells by Drp1 S637A overexpression (Fig. 6b). Mitochondrial mass, as indicated by CypD, cytochrome *C* and Tom20 in western blot, was also rescued by Drp1 S637A overexpression (Fig. 6c, Supplementary Fig. 4b–d). Consistently, AMPK and TSC2 phosphorylation was revered, and S6 phosphorylation was inhibited in *PGAM5*$^{-/-}$ ARPE-19 cells by Drp1 S637A overexpression (Fig. 6d, Supplementary Fig. 4e–g). As to ATP level, short-term Drp1 overexpression decreased ATP level in *PGAM5*$^{-/-}$ to the level similar to WT cells (Fig. 6e). When cells were cultured for 8 weeks, Drp1 S637A overexpression partially restored the decreased ATP level in *PGAM5*$^{-/-}$ cells (Fig. 6f). Cell senescence in *PGAM5*$^{-/-}$ cells, as indicated by SA-βGal

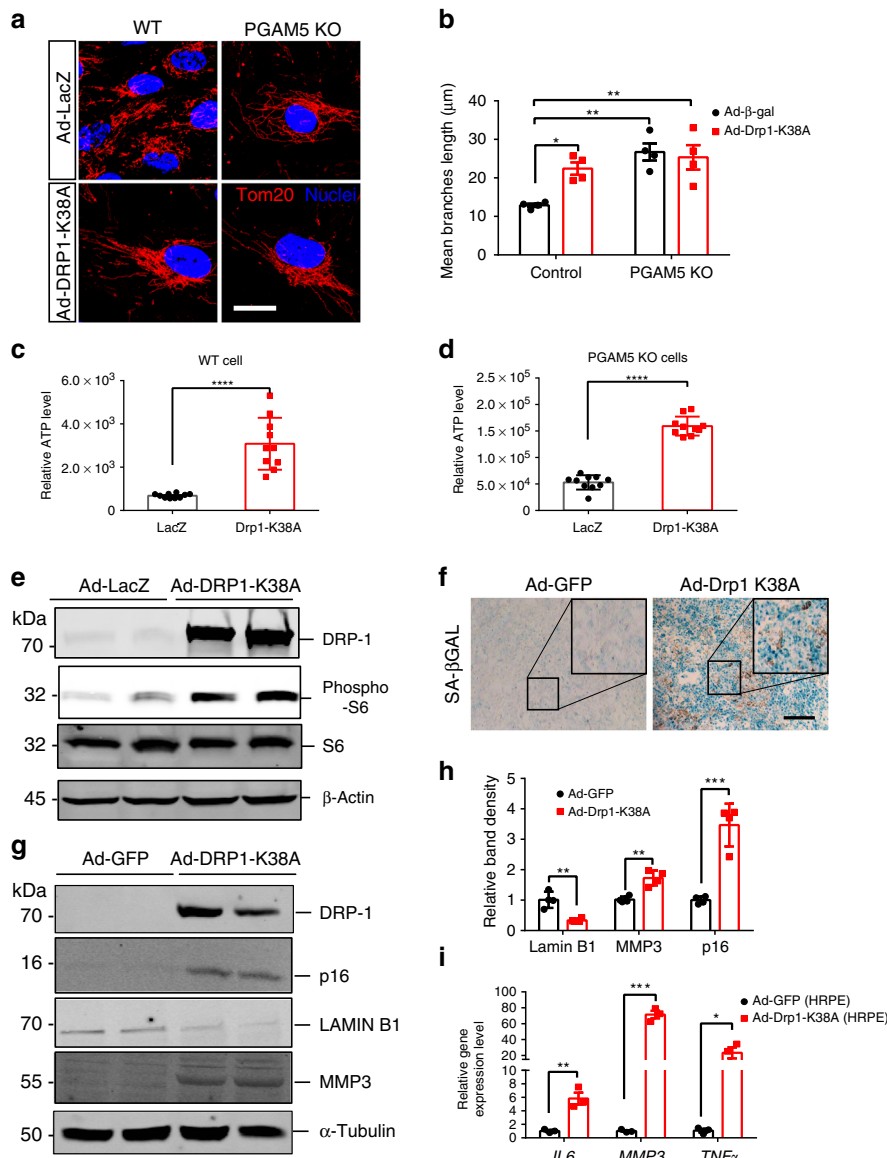

**Fig. 5 Drp1-K38A overexpression phenocopies *PGAM5*−/− mitochondrial phenotype and cell senescence. a** Tom20 immunostaining in WT and *PGAM5*−/− ARPE-19 cells infected with adenovirus expressing Ad-LacZ or Drp1-K38A mutant, showing that Ad-Drp1-K38A mutant mimics *PGAM5*−/− hyperfusion phenotype. Scale bar = 20 μm. $n = 4$. **b** Quantification of mitochondrial branches in **a** using ImageJ. $n = 4$, *$p < 0.05$, **$p < 0.005$, analyzed by two-way ANOVA Tukey's multiple comparisons test. Error bars, mean ± s.e.m. **c** ATP production in WT ARPE-19 cells overexpressing Ad-LacZ or Ad-Drp1-K38A. $n = 10$ biologically independent samples, ****$p < 0.0001$, two-tailed unpaired *t*-tests, error bars, mean ± s.d. **d** ATP production in *PGAM5* KO cells overexpressing Ad-LacZ or Ad-Drp1-K38A. $n = 10$ biologically independent samples, ****$p < 0.0001$, two-tailed unpaired *t*-tests, error bars, mean ± s.d. **e** Expression of DRP1, phosphor-S6 and S6 in Ad-LacZ or Ad-Drp1-K38A overexpressing HRPE cells at 7 days after infection. Samples were collected at 48 h after the last medium change for western blot. β-Actin was used as loading control. $n = 3$. **f** SA-gal staining of HRPE cells 5 weeks after Ad-GFP or Ad-Drp1-K38A infection (MOI = 100). Scale bar = 500 μm. Boxed region represents the magnified picture in the figure. $n = 3$. **g** Expression of Drp1, Lamin B1, MMP3 and P16Ink4a protein at 8 weeks after infection. Samples were collected at 48 h after the last medium change for western blot. α-Tubulin was used as loading control. $n = 4$. **h** Quantification of bands in **g**. $n = 4$, **$p = 0.0018$ (Lamin B1), **$p = 0.0026$ (MMP3), ***$p = 0.0004$, two-tailed unpaired *t*-tests, error bars, mean ± s.d. **i** mRNA level of *IL6*, *MMP3* and *TNFα* were measured by qRT-PCR at 8 weeks after infection as shown in **f**. Samples were collected at 48 h after the last medium change. $n = 3$, *$p = 0.0338$, **$p = 0.0057$, ***$p = 0.0001$, two-tailed unpaired *t*-tests, error bars, mean ± s.e.m. For assays in the figure, *n* represents the number of biologically independent experiments unless otherwise specified. Images were captured under same settings, and representative images were shown. Source data are available as a Source Data file.

activity, was also partially rescued by Drp1 S637A overexpression (Fig. 6g, h).

**Age-dependent response to oxidative stress by *PGAM5* deletion.** *PGAM5*−/− lymphocytes and MEF cells are resistant to oxidative stress in caspase-independent manner[39]. To address whether *PGAM5* deletion protects RPE cells from oxidative stress, we employed a NaIO3-driven RPE degeneration model[40]. Fundus pictures showed no obvious difference between *Pgam5*−/− and WT mice at 2 or 18 months (Fig. 7a, f). However, at 5 days after NaIO3 injury, the number of white spots, indicative of RPE degeneration, appeared to be less in 2-month-old *Pgam5*−/− fundi compared to WT mice, suggesting *Pgam5* deletion renders oxidative stress resistance (Fig. 7b). Protection of RPE layer was

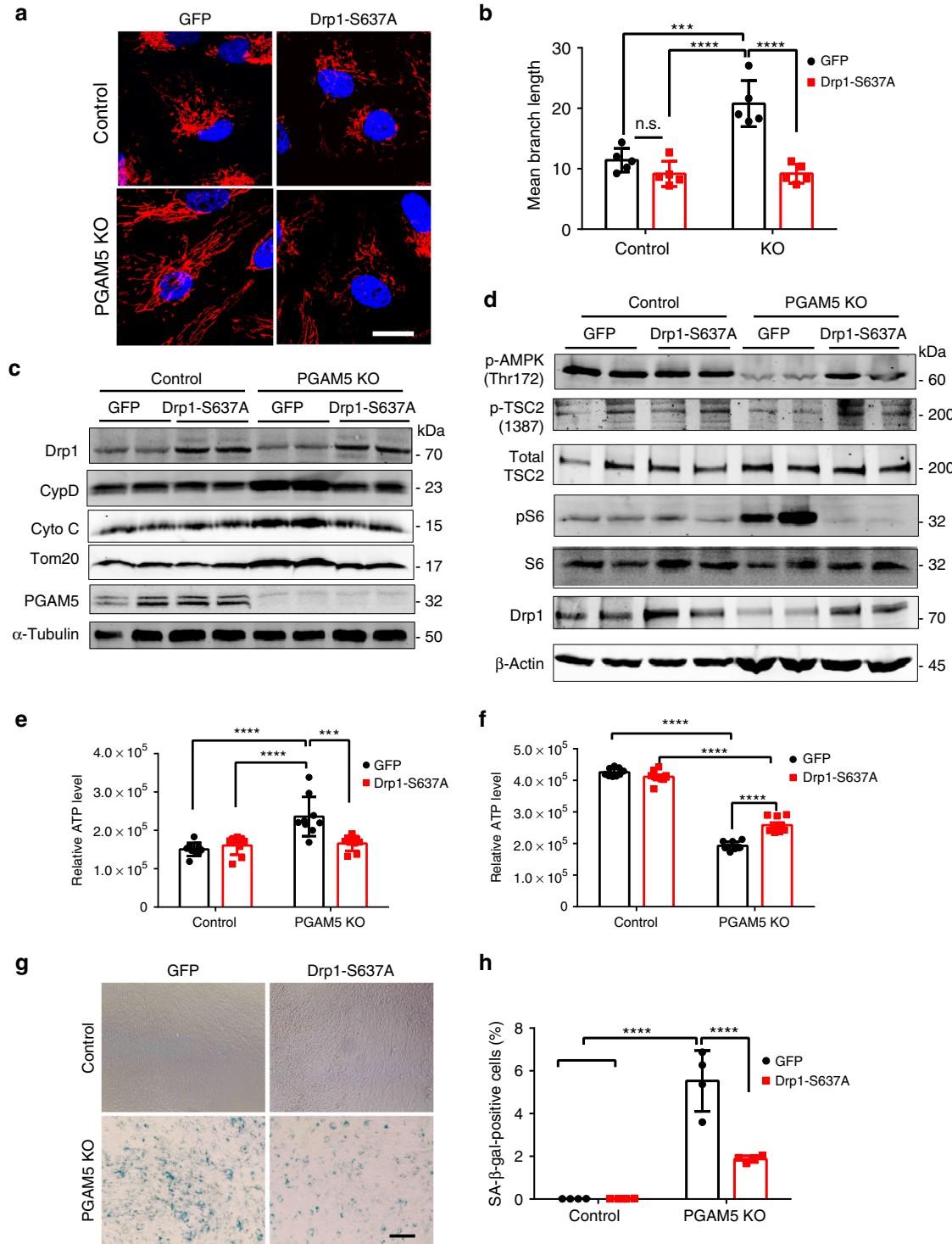

confirmed by ZO-1 staining of RPE flatmount and histological sections (Fig. 7c–e). The damage induced by NaIO₃-derived oxidants depends on its local concentration after tail vein injection. The ocular arteries and veins converge through the optic disc in the retina, generating a concentration gradient of NaIO₃ with higher concentration adjacent to the optic disc. By ZO-1 staining, the tight junction of RPE cells were interrupted in WT mice (Fig. 7d, left panel), as confirmed by the dislocated RPE cells from the RPE layer in histological sections (Fig. 7e, WT panels). However, in *Pgam5⁻/⁻* mice, RPE damage was much less severe in the middle and periphery region compared to the

corresponding regions in the WT as shown by ZO-1 (Fig. 7d, right panel) and histology staining (Fig. 7e, KO middle and periphery panels). However, when similar experiments were performed in the 18-months-old mice, RPE protection rendered by *Pgam5* deletion was lost (Fig. 7f–j).

To confirm the results in vitro, neutral red assay and propidium iodide (PI) staining were used to measure RPE cell viability after NaIO₃ treatment. *PGAM5⁻/⁻* ARPE-19 cells showed significantly better viability than controls at 12 h after NaIO3 treatment at several concentrations (Fig. 8a). PI staining confirmed less cell death in *PGAM5⁻/⁻* cells compared to

**Fig. 6 Rescue of $PGAM5^{-/-}$ accelerated senescence phenotype by Drp-S637A overexpression. a** Mitochondrial morphology outlined by Tom20 antibodies in WT and $PGAM5^{-/-}$ ARPE-19 cells overexpressing GFP or DRP1 S637A, respectively. Scale bar = 20 μm. $n = 5$. **b** Quantification of mitochondrial branch in **a**. $n = 5$, ***$p = 0.0007$, ****$p < 0.0001$, n.s. no significant. two-way ANOVA Tukey's multiple comparisons test. Error bars, mean ± s.d. **c** Upregulation of mitochondrial proteins (Tom20, CYTC, CYPD) were rescued by Drp1 S637A overexpression in $PGAM5^{-/-}$ ARPE-19 cells. α-Tubulin was used as loading control. $n = 4$. **d** Expression of phosphor-AMPK, phosphor-TSC2, total-TSC2, phosphor-S6, total S6 and Drp1 in WT and $PGAM5^{-/-}$ ARPE-19 cells transduced with GFP or Drp1 S637A expressing lentivirus. β-Actin was used as loading control. $n = 4$. **e** ATP level in short-term (1 week) culture of WT and $PGAM5^{-/-}$ ARPE-19 cells transduced with GFP or Drp1 S637A expressing lentivirus. $n = 9$ biologically independent samples. ***$p = 0.0005$, ****$p < 0.0001$, two-way ANOVA Tukey's multiple comparisons test. Error bars, mean ± s.d. **f** ATP level in long-term (8 week) culture of WT and $PGAM5^{-/-}$ ARPE-19 cells transduced with GFP or Drp1 S637A expressing lentivirus. $n = 9$ biologically independent samples. ****$p < 0.0001$, two-way ANOVA Tukey's multiple comparisons test. Error bars, mean ± s.d. **g** Partial rescue of senescence phenotype in $PGAM5^{-/-}$ ARPE-19 cells by Drp1 S637A overexpression as shown by SA-β-Gal assay. Scale bar = 100 μm. $n = 4$. **h** Quantification β-gal-positive cells in **g**. $n = 4$. ****$p < 0.0001$, two-way ANOVA Tukey's multiple comparisons test. Error bars, mean ± s.d. For assays in the figure, $n$ represents the number of biologically independent experiments unless otherwise specified. Images were captured under same settings, and representative images were shown. Source data are available as a Source Data file.

controls (Fig. 8b, c). To further investigate the underlying mechanism, qRT-PCR was used to evaluate the expression of a list of anti-oxidative genes. The expression of $FOXO4$, $SOD1$ and $NRF2$ was significantly upregulated in $PGAM5^{-/-}$ ARPE-19 cells, consistent with the observed ROS level increase (Fig. 8d). Increased NRF2 protein level was confirmed by western blot analysis (Fig. 8f, g). However, the upregulation was lost when cells were cultured for 8 weeks. Decreased expression of $FoxO4$, $Sod1$ and $Nrf2$ mRNA and Nrf2 protein expression was detected instead (Fig. 8e, h, i). Consistently, upregulation of anti-oxidative genes was also confirmed in the RPE/choroid of 2-months-old $Pgam5^{-/-}$ mice (Supplementary Fig. 5a). Moreover, the upregulation of $Nrf2$ and its downstream gene $Nqo1$ was lost in the RPE/choroid tissue in 18-months-old $Pgam5^{-/-}$ mice (Fig. 8j, Supplementary Fig. 5b). These suggest that $PGAM5$ deletion renders age-related resistance to oxidative stress.

**Loss of $PGAM5$ promotes immune regulatory pathway.** Interferon regulatory factor (IRF) pathway is critical in innate immune response activation, inflammation and senescence[41,42]. As phospho-rH2AX, a marker for DNA damage, was activated by PGAM5 deletion, we asked whether IRF pathway is activated in $PGAM5^{-/-}$ cells. By western blot, IRF-3 phosphorylation is significantly upregulated in $PGAM5^{-/-}$ ARPE-19 cells (Fig. 9a). Consistently, RNA level of $IFN\beta$, a downstream gene of IRF-3, was significantly upregulated in $PGAM5^{-/-}$ or PGAM5 silenced ARPE-19 cells (Fig. 9b, c). Moreover, $Ifn\beta$, $Ccl2$ and $Ccl4$ RNA level was increased during aging in RPE/choroid tissue, and a trend of higher expression was observed in $Pgam5^{-/-}$ mice (Fig. 9d and Supplementary Fig. 5c, d). These suggest the activation of IRF pathway by PGAM5 deletion. To examine its consequence, subretinal microglia cells were quantified after Iba-1 staining. As shown in Fig. 9e, f, subretinal microglia cells were increased during aging, and higher numbers were observed in $Pgam5^{-/-}$ mice compared to the WT controls. Taken together, we found that PGAM5 deficiency promotes the IRF/IFN-β pathway activation, which results in more microglia accumulation in the RPE/subretinal space.

## Discussion

Our results reveal an important function for PGAM5 in senescence. We found that: (1) $PGAM5$ deletion leads to accelerated senescence in vitro and in mice; (2) $PGAM5$ deletion leads to mitochondrial hyperfusion and less mitochondrial turnover; (3) PGAM5 deletion leads to activation of mTOR and IRF/IFN-β pathways, which are essential pathways for senescence; (4) Drp1-K38A overexpression phenocopies but Drp1-S637A overexpression rescues the mitochondrial morphology change, mTOR

activation and senescence in $PGAM5^{-/-}$ RPE cells; (5) PGAM5 deletion renders age-dependent RPE cells resistance to oxidative stress. These data establish a link between PGAM5, mitochondrial fission, senescence and anti-oxidative response, reinforcing the importance of mitochondrial dynamics in regulating cellular senescence (Fig. 10).

Accumulation of senescent cells has been observed during aging and age-related diseases[43]. Elimination of senescent cells could restore tissue homeostasis or prevent neurodegeneration, inciting a senolytic approach for treating age-related diseases[44,45]. The molecular mechanism underlying senescence remains unclear, but ROS, mTORC1 and IRF/IFN-β each was shown to contribute to cell senescence[5,41,46]. We found that PGAM5 deletion leads to accelerated senescence, as indicated by increased cell volume, senescence β-gal staining, and a series of other markers and SASPs. $PGAM5^{-/-}$ mice show reduced survival by 18-month, with ~50% of them showing hunchback, swollen foot and odd gait phenotypes, and Parkinson's-like movement disorders as reported previously[16]. This phenotype was confirmed by senescence biomarker staining. In contrast to WT mice, 18-months-old $Pgam5^{-/-}$ mice show increased P16$^{Ink4a}$, phospho-rH2AX staining, p53 and MMP3, decreased Lamin B1, as well as increased age-related accumulation of microglial cells in the subretinal space. These support $Pgam5^{-/-}$ mouse as model for accelerated senescence. As Parkinson's disease is age-related, linking senescence to Parkinson-like syndrome in vivo supports senescence could contribute to Parkinson's disease[47]. Several mouse models of accelerated cellular senescence have been reported, confirming the important of genome instability ($Ercc1^{-/-}$ mouse), ROS ($SOD1^{-/-}$ mouse) and nuclear membrane compromise ($Lmna$ mouse) in senescence[48-52]. Our $Pgam5^{-/-}$ mice represent an accelerated senescence mouse model caused by defective mitochondrial dynamics. Of note, accumulation of dysfunctional mitochondria is associated with senescence or age-related diseases (16–18). Our model represents a model more related to mitochondrial metabolism and natural aging process, underscoring the important of mitochondrial dynamics in normal life, senescence and aging, and potentially age-related diseases. We explored whether PGAM5 expression level is altered in healthy old people or AMD patients. Analysis of PGAM5 expression in GSE50195 (ref. [53]) and GSE29801 (ref. [54]) revealed $PGAM5$ mRNA level is not significantly altered with age or in AMD disease (Supplementary Fig. 6a–c). Further studies would focus on total and cleaved PGAM5 protein level and $PGAM5$ mutations in age-related disease models. Alternatively, PGAM5 activity might not change during aging, but still mediate multiple pathways involved in senescence.

The action of PGAM5 may reflect on multiple mechanisms: (1) We confirm in vitro that cleaved PGAM5 functions as a

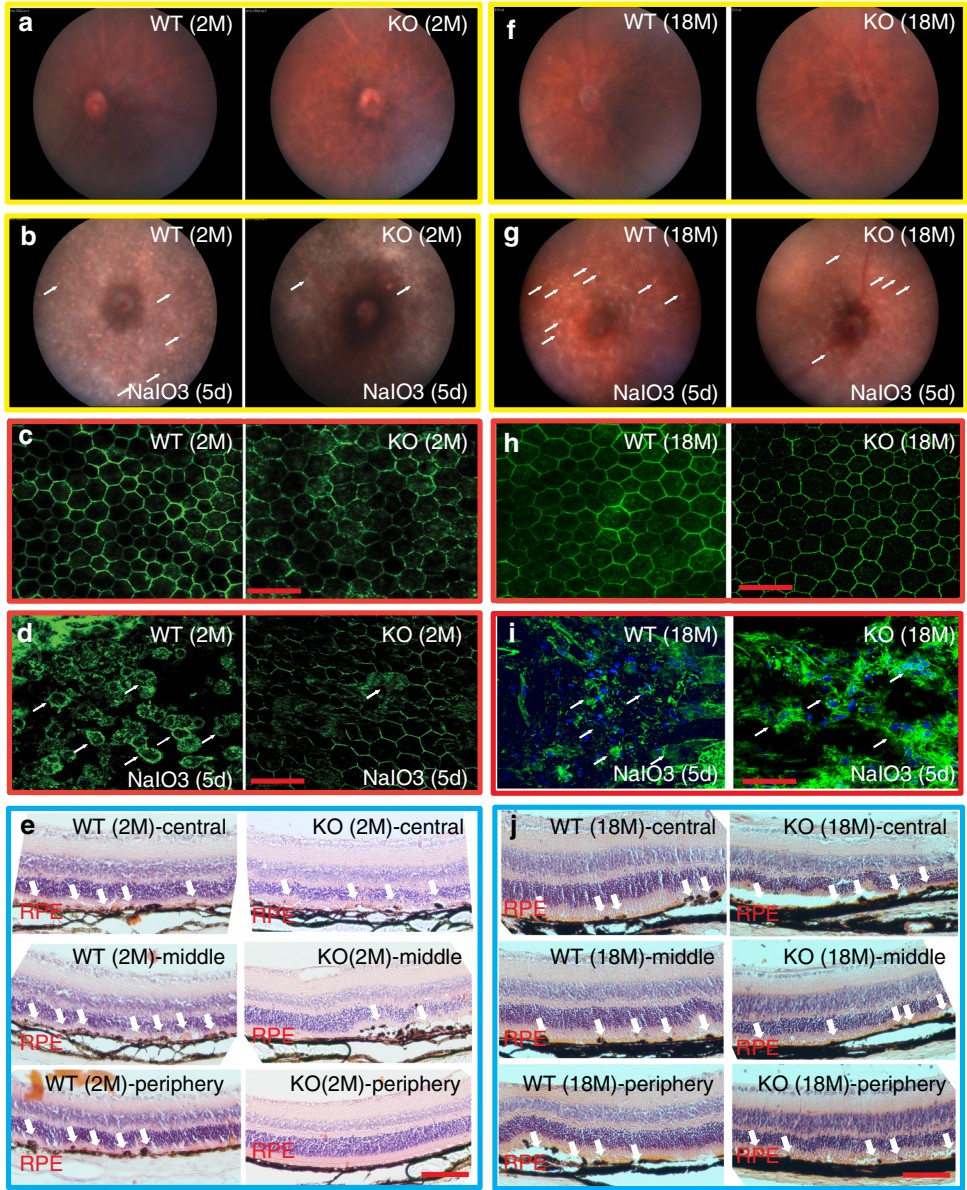

**Fig. 7 Pgam5 deletion protects RPE cells from oxidative stress in young but not old mice. a** Normal fundus in 2-months-old WT (left panel) and *Pgam5*−/− (right panel) mice. **b** Fundus of 2-months-old WT (left panel) and *Pgam5*−/− (right panel) mice at 5 days after NaIO₃ injection. Arrows point to the "white spots" that represent degenerated RPE. **c** ZO-1 staining of the retinal flatmount in 2-months-old WT (left panel) and *Pgam5*−/− (right panel) mice. Scale bar = 50 μm. **d** ZO-1 staining of the retinal flatmount in 2-months-old WT (left panel) and *Pgam5*−/− (right panel) mice at 5 days after NaIO3 injection. Scale bar = 50 μm. Arrows point to the disrupted RPE regions. **e** Representative H&E staining of sections from 2-months-old WT and *Pgam5*−/− mice at 5 days after 20 mg/kg NaIO₃ injection. Top, middle and down panels represent the central, middle and periphery regions of the retina, respectively. Scale bar = 100 μm. Arrows point to disrupted RPE regions. **f** Normal fundus in 18-months-old WT (left panel) and *Pgam5*−/− (right panel) mice. **g** Fundus of 18-months-old WT (left panel) and *Pgam5*−/− (right panel) mice at 5 days after NaIO₃ injection. Arrows point to the "white spots" that represent degenerated RPE. **h** ZO-1 staining of the retinal flatmounts in 18-months-old WT (left panel) and *Pgam5*−/− (right panel) mice. Scale bar = 50 μm. **i** ZO-1 staining of the retinal flatmount in 18-months-old WT (left panel) and *Pgam5*−/− (right panel) mice at 5 days after NaIO3 injection. Scale bar = 50 μm. Arrows point to the disrupted RPE regions. **j** Representative H&E staining of sections from 18-months-old WT and *Pgam5*−/− mice at 5 days after 20 mg/kg NaIO₃ injection. Top, middle and down panels represent the central, middle and periphery regions of the retina, respectively. Scale bar = 100 μm. Arrows point to disrupted RPE regions. *n* = 4 biologically independent experiments for each group in this figure. Images were captured under same settings, and representative images were shown.

phosphatase to dephosphorylate Drp1 mediated by Axin1. Damaged mitochondria (such as CCCP treatment) induces PGAM5 cleavage and release to cytosol, where it binds to Axin1. Drp1 (Ser-637) is also recruited to Axin1 and subsequently dephosphorylated by PGAM5. The dephosphorylated Drp1 binds to mitochondria and promotes mitochondrial fission and turnover, therefore removing the damaged mitochondria. PGAM5

deletion leads to accumulation of phosphorylated and therefore inactive form of Drp1. This leads to impaired mitochondrial fission and less mitochondrial turnover, which generated more tubular mitochondria. The change in mitochondrial dynamics has two consequences: ROS increase and higher ATP production initially but lower ATP level over longer time. Initial higher ATP production may be good for cells energy-wise, but it stimulates

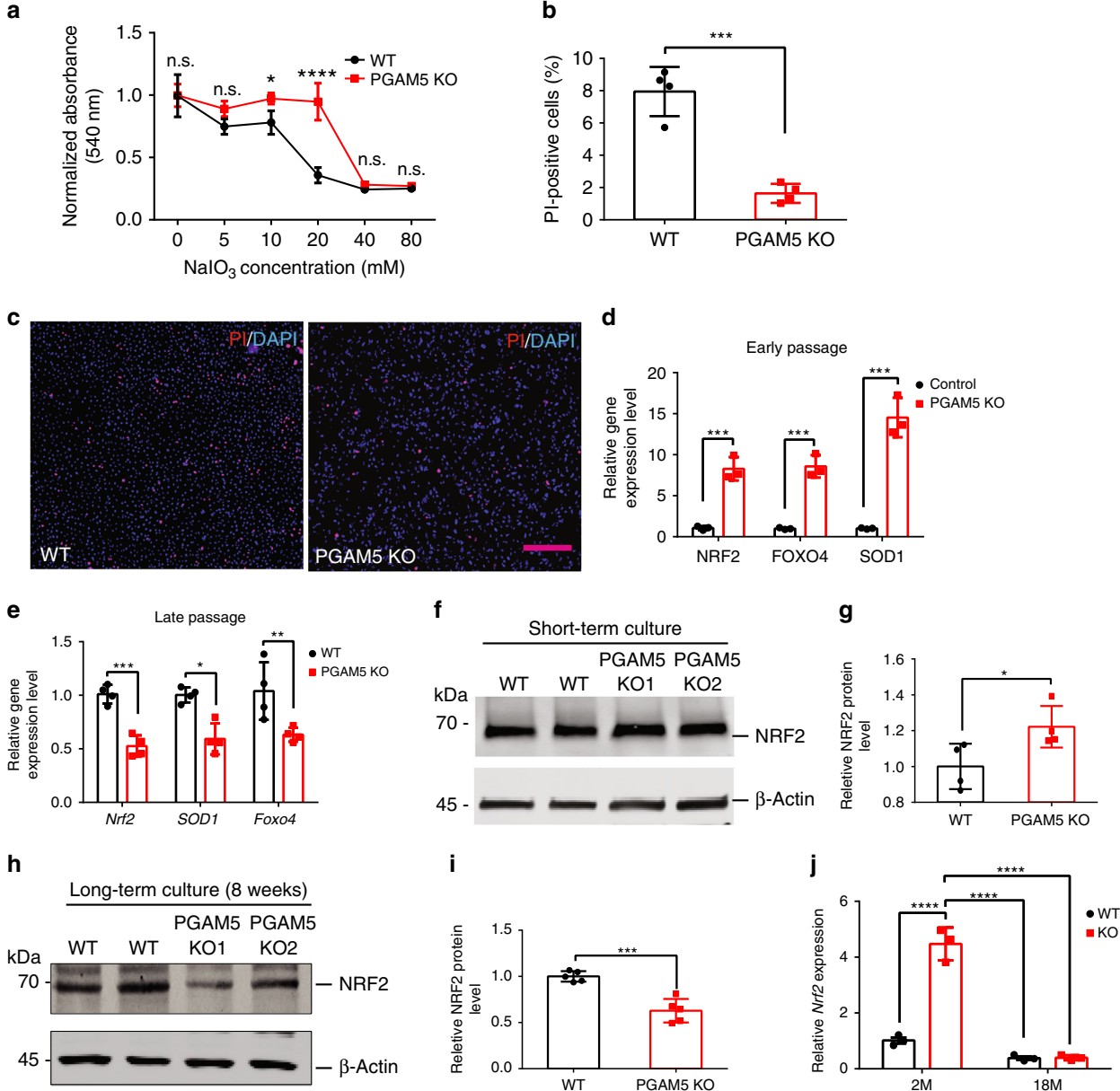

**Fig. 8 Age-dependent regulation of anti-oxidative genes by PGAM5. a** Cell survival by neutral red assay in WT and *PGAM5$^{-/-}$* ARPE-19 cells after overnight NaIO$_3$ treatment using indicated concentrations. $n = 4$, *$p < 0.05$, ****$p < 0.0001$, two-way ANOVA Tukey's multiple comparisons test. Error bars, mean ± s.d. **b** Quantification of PI staining showing cell death in WT and *PGAM5$^{-/-}$* ARPE-19 cells after overnight NaIO$_3$ (10 mM) treatment using Nikon Elements software and ImageJ. $n = 4$, ***$p = 0.0002$, two-tailed unpaired *t*-tests, error bars, mean ± s.d. **c** Representative PI staining in **b**. Scale bar = 250 μm. $n = 4$. **d** Upregulation of anti-oxidative genes (*NRF2*, *FOXO4* and *SOD1*) in *PGAM5$^{-/-}$* ARPE-19 cells compared to control cells after short-term (1 week) culture. Data were normalized to GAPDH. $n = 3$, ***$p = 0.001$(*Nrf2*), $p = 0.0007$(*Foxo4*), $p = 0.0006$, two-tailed unpaired *t*-tests, error bars, mean ± s.d. **e** Downregulation of anti-oxidative genes (*NRF2*, *FOXO4* and *SOD1*) in *PGAM5$^{-/-}$* ARPE-19 cells compared to control cells after long-term (8 weeks) culture. Data were normalized to GAPDH. $n = 4$, ***$p = 0.0003$; **$p = 0.0022$; *$p = 0.0258$, two-tailed unpaired *t*-tests, error bars, mean ± s.d. **f** Upregulation of NRF2 protein in *PGAM5$^{-/-}$* ARPE-19 cells compared to control cells after short-term (1 week) culture. Data were normalized to β-actin. $n = 4$. **g** Quantification of data in **f**. $n = 4$, *$p = 0.042$, two-tailed unpaired *t*-tests, error bars, mean ± s.d. **h** Downregulation of NRF2 protein in *PGAM5$^{-/-}$* ARPE-19 cells compared to control cells after long-term (8 weeks) culture. Data were normalized to β-actin. $n = 5$. **i** Quantification of data in **h**. $n = 5$, ***$p = 0.0003$, two-tailed unpaired *t*-tests, error bars, mean ± s.d. **j** *Nrf2* expression as measured by qRT-PCR in the RPE/choroid of WT and *Pgam5$^{-/-}$* mice at 2 and 18-months old. Data were normalized to *Gapdh*. $n = 3$, ****$p < 0.0001$, two-way ANOVA Tukey's multiple comparisons test. Error bars, mean ± s.d. For assays in the figure, $n$ represents the number of biologically independent experiments unless otherwise specified. Images were captured under same settings, and representative images were shown. Source data are available as a Source Data file.

mTORC1 via AMPK-TSC2 pathway, which leads to senescence, eventually low ATP production. Consistently, we found that overexpression of Drp1-K38A mutant, an inactivated form of Drp1, phenocopies the ATP upregulation, mTOR activation and senescence caused by PGAM5 deletion. Drp1-S637A mutant, a constitutively active form of Drp1, rescues the marker expression and phenotypes above in *PGAM5$^{-/-}$* cells. These establish a causal relationship between mitochondrial hyperfusion and senescence, and a critical role for Drp1 in mediating PGAM5 function in senescence. ROS increase in *PGAM5$^{-/-}$* cells does not

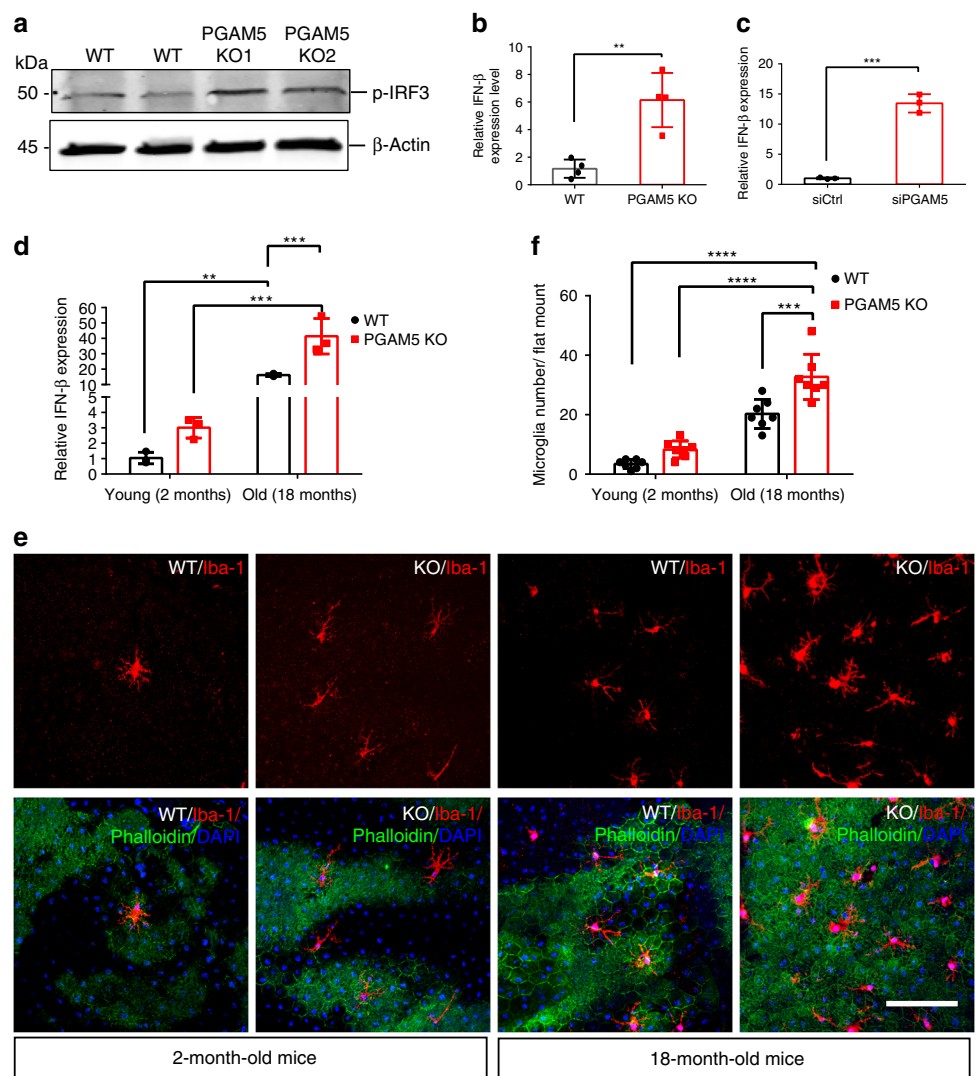

**Fig. 9 *PGAM5* deletion promotes the IRF/IFN-β pathway. a** Upregulation of phosphor-IRF-3 protein by western blot in *PGAM5⁻/⁻* ARPE-19 cells compared to control cells. β-Actin was used as loading control. $n = 3$. **b** Upregulation of *IFN-β* by qRT-PCR in *PGAM5⁻/⁻* ARPE-19 cells compared to control cells. Data were normalized to GAPDH. $n = 4$, ***$p = 0.003$, two-tailed unpaired *t*-tests, error bars, mean ± s.d. **c** Upregulation of *IFN-β* by qRT-PCR in ARPE-19 cells after PGAM5 silencing by siRNA compared to control cells. Data were normalized to *GAPDH*. $n = 3$, ***$p = 0.003$, two-tailed unpaired *t*-tests, error bars, mean ± s.d. **d** Upregulation of *Ifn-β* by qRT-PCR in the RPE/choroid of *Pgam5⁻/⁻* mice aged at 2 and 18-months old. $n = 3$, **$p = 0.0085$, ****$p < 0.0001$, two-way ANOVA Tukey's multiple comparisons test. Error bars, mean ± s.d. **e** Representative images showing microglia in subretinal space in 2 and 18-months-old WT and *Pgam5⁻/⁻* mice by Iba-1 staining. Scale bar = 100 μm. $n = 7$. **f** Quantification of microglia cell number per flatmount in **e**. $n = 7$, ***$p < 0.001$, ****$p < 0.0001$, two-way ANOVA Tukey's multiple comparisons test. Error bars, mean ± s.d. For assays in the figure, *n* represents the number of biologically independent experiments. Images were captured under same settings, and representative images were shown. Source data are available as a Source Data file.

affect mTOR pathway activation in our study but could contribute to senescence through other mechanisms[5]; (2) PGAM5 may regulate mitochondrial biogenesis. We found that PGAM5 deletion leads to drastic downregulation of PGC1α, which is a critical co-transcriptional factor for mitochondrial biogenesis[55]. Consistently, cleaved PGAM5 has been shown to be released to the cytoplasm and activate mitochondrial biogenesis[18]. A decrease in mitochondrial biogenesis could contribute to less mitochondrial turnover, leading to accumulation of damaged or senescent mitochondria in the long run. Further experiments would directly confirm mitochondrial biogenesis regulated by PGAM5; (3) PGAM5 deletion leads to IRF/IFN-β pathway activation. Activation of IFNβ by either PGAM5 deletion or PGAM5 siRNA suggests PGAM5 deletion could lead to IRF/IFN-β pathway activation. Consistently, we observed increased age-related accumulation of microglial cells

in the subretinal space in *Pgam5⁻/⁻* mice. We propose that mitochondrial or nuclear DNA damage caused by *PGAM5* deletion could lead to IRF/IFN-β pathway activation. In total, we found ROS increase, mTOR and IRF/IFN-β activation in *PGAM5⁻/⁻* cells. Therefore, our study provides molecular underpinning leading to ROS elevation, and mTOR and IRF/IFN-β activation, which eventually result in senescence. Besides Drp1, other PGAM5 substrates could also contribute to PGAM5 function. For example, PGAM5 may also directly regulate mitophagy and autophagy through PINK1 and FUNDC1. Future work is needed to dissect the full spectrum of PGAM5 targets and function.

Controversies exist regarding mitochondrial dynamics, especially mitochondrial fusion/fission in senescence. We found that PGAM5 deletion leads to mitochondrial hyperfusion, less mitochondrial turnover, and eventually accelerated senescence in

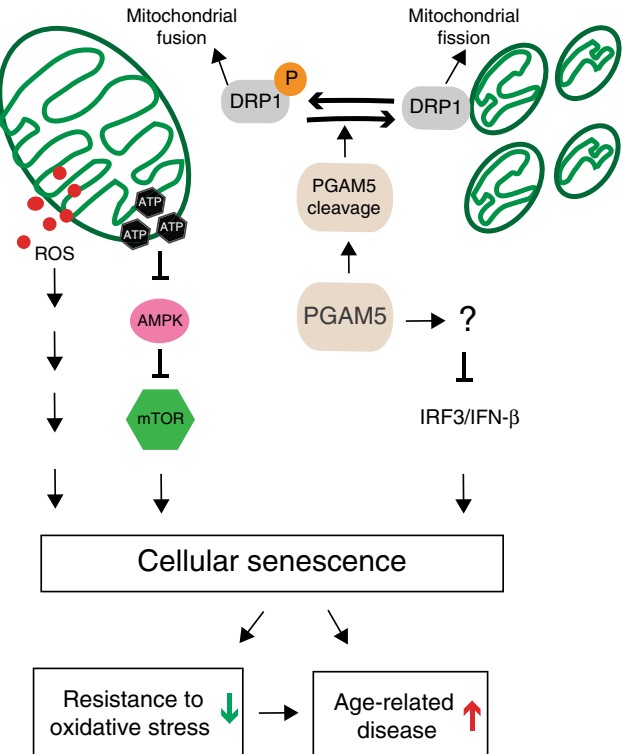

**Fig. 10 Proposed model for PGAM5 in senescence.** In this model, PGAM5 regulates senescence through several mechanisms. First, *PGAM5* is required for regulating mitochondrial dynamics. PGAM5 deletion leads to elevation of Drp1 phosphorylation at Ser-637, which tip the balance of mitochondrial dynamics toward hyperfusion and less mitochondrial turnover. As result, this results in elevated ATP and ROS production, activation of AMPK–mTOR pathway, and accelerated senescence. Second, PGAM5 deletion promotes the IRF/IFN-β pathway, which could also contribute to cell senescence. PGAM5 deletion leads to resistance to oxidative stress, possibly through mitochondrial hyperfusion and/or upregulation of anti-oxidative genes. This resistance is lost in aging possibly due to the accelerated senescence caused by PGAM5 deletion. Our model highlights a critical role for PGAM5 in regulating mitochondrial dynamics, senescence and age-related oxidative response.

mice. Consistent with our observation, Lee et al.[56] reported that knocking out hFis1 (Mitochondrial fission 1 protein, which recruits Drp1 to mitochondrial surface for fission) generates elongation of mitochondria and cellular senescence phenotypes in vitro. Rana et al.[8] showed that upregulated Drp1 expression in midlife in Drosophila lead to more mitochondrial fission and prolonged lifespan. However, contradictory results also exist. Scheckhuber et al.[57] found that reducing mitochondrial fission by deleting Dnm1p (Drp1 homologous gene) in fungal models leads to fungal with low growth rate but extended lifespan. Maintenance of the fused mitochondrial network in *C. elegans* is necessary for longevity, and absence of mitochondrial fusion in the Deltamgm1 mutant leads to a striking reduction of both replicative and chronological lifespan in *C. elegans*[7,58]. In a recent study, disruption of either mitochondrial fission or fusion significantly reduces medium lifespan in *C. elegans*[9]. For mouse study, Nishimura et al.[59] reported that Drp1 and filamin interaction induced by hypoxia could induce mitochondrial hyperfission and myocardial senescence. The different outcomes from these studies may reflect on the lifespan and other characteristics of different species. For example, single cell organism may have different response to mitochondrial fission/fusion from mammalians. Extreme conditions like hypoxia may induce acute mitochondrial responses, therefore short-term and long-term responses could be different.

Our study also sheds light on the relationship between senescence and anti-oxidative response. It is established that increased ROS and oxidative stress can lead to senescence and age-related disease. Aging (or senescent) cells could also have reduced anti-oxidative capabilities. For example, aging RPE cells have been shown to be more vulnerable to oxidative damage due to impaired to Nrf2 signaling[60]. Our results show that young *Pgam5⁻/⁻* mice are more resistant to NaIO3 induced RPE damage compared to control WT mice. This phenotype correlates the upregulation of anti-oxidative genes *NRF2*, *SOD1* and *FOXO4* in short-term cultured *PGAM5⁻/⁻* cells. Upregulation of NRF2 in short-term cultured RPE *PGAM5⁻/⁻* cells is consistent with PGAM5's function in forming PGAM5/KEAP1/NRF2 complex for NRF2 degradation[19]. Regulation of FOXO4 by PGAM5 is also consistent with a recent finding that PGAM5 promotes lasting FoxO activation after developmental mitochondrial stress and extends lifespan in *Drosophila*[61]. Several groups also verified PGAM5 deficiency renders cells resistance to oxidative stress[14,39]. However, we found by surprise that the anti-oxidative resistance was lost in 18-month-old *Pgam5⁻/⁻* mice, consistent with the failure to upregulate NRF2 in long-term cultured *PGAM5⁻/⁻* cells and in vivo. These findings suggest that senescence may change the landscape of anti-oxidative response of the cells, and the beneficial anti-oxidative function by PGAM5 deletion is lost due to its role in senescence. Our study also exemplifies nature's balance on short-term benefit to long-term undesired effects, which draws caution to correlate short-term mechanistic studies with long-term phenotypic analyses. Further work is needed to decipher the role and mechanism of senescence in anti-oxidative response.

Our findings that PGAM5 regulates mitochondrial dynamics, senescence and age-related anti-oxidative response could have therapeutic implications. Harnessing the effect of PGAM5 deletion in anti-oxidation but preventing its undesired effect in senescence could be beneficial in age-related diseases, including Parkinson's disease and AMD.

## Methods

**Generation of *Pgam5⁻/⁻* mice and animal experiments**. Animal studies were conducted in accordance with the ARVO statement for the Use of Animals in Ophthalmic and Vision Research and were approved by the Institutional Animal Care and Use Committees at the Tulane University. Validated *Pgam5* gene-targeted ES cells (C57BL/6N) (MGI ID: 4432458) were purchased from the European Mouse Mutant Cell Repository. The ES cells were injected into blastocyst donors to generate chimeric mice. After confirmation of germ line transmission, *Pgam5 flox* sperms were used to re-derive mice into C57BL/6J background at Charles River. *Pgam5* flox/flox mice were then crossed with CAG-Cre mice to generate *Pgam5⁻/⁻* mice. The genotyping primers for *PGAM5* genotyping include: primer 1: GGCC TGTCTTATCACCAGCAGAAGTA. Primer 2: GGAGACATTGTGACCATCCAA CTCT. Primer 3: CTTGTTGATATCGTGGTATCGTT. Primer 4: CTTCACCAG CCTGTCTTCACCA. Primer 5: TGGGAAGCAAGGCAGGG. (Expected band size: WT: primer 1 and 2: 1 kb, primer 4 and 5: 560 bp; KO: primer 1 and 2: 300 bp, primer 4 and 5: no band; Flox: primer 1 and 2: no band, primer 1 and 3: 290 bp, primer 4 and 5: 560 bp). The primers for testing the absence of the rd8 background include: *mCrb1* mF1: GTGAAGACAGCTACAGTTCTGATC; *mCrb1mF2: GCCC CTGTTTGCATGGAGGAAACT GGAAGACAGCTACAGTTCTTCTG; mCrb1mR*: GCCCCATTTGCACACTGATGAC[62]. For NAIO3 injury experiments, 20 mg/kg NaIO₃ (in 80 μl) was administrated into 2-months-old or 18-months-old mice intravenously. Fundus images were captured at 5 days post injection using Micron III (Phoenix). The mice were then killed, and the eye balls were fixed in 4% PFA for histology or flatmount ZO-1 staining.

**_PGAM5⁻/⁻_ ARPE-19 cell line generation using CRISPR/Cas9 technology.** *PGAM5⁻/⁻* ARPE-19 cell lines were generated according to ref. [63]. Briefly, lenti-guide puro vector (addgene: #52963)[64] was used to insert guide RNA targeting human *PGAM5* gene after BsmBI (Fermentas) digestion of the vector. Two different guide RNAs were designed online (https://zlab.bio/guide-design-resources) (Supplementary Table 1). The ligated and empty control vectors was packaged into lentiviral vectors and transduced into ARPE-19 cells, respectively. After screening

by puromycin (2 µg/ml) for 1 week, Ad5-CMV-Cas9 (Cat #: VVC-U of Iowa-4683), an adenovirus vector expressing Cas9 gene, was added into cells at 100:1 MOI. After 4 days, cells were examined by western blot for knockout efficiency. Two independent lines have been established with similar knockout efficiency and phenotypes.

**Adenovirus and lentivirus generation**. Adenoviruses were generated as described[65]. Drp1 K38A mutant was amplified from pCDNA3-Drp1 K38A mutation vectors (Gifts from Dr. Wang Wang at University of Washington) respectively by using primers: Drp1-up: 5′-AATGGTACCATGGAGGCGCTAATTCCT-3′ and Drp1-down: 5′- AACTCGAGTCACCAAAGATGAGTCTCCCGGA-3′. After digestion by *Kpn*I and *Xho*I, they were subcloned into pShuttle-CVM vector (Ad-EASY system). The positive clones were cut with *Pme*I and transformed into *E. coli* with adenovirus vector for recombination. Positive clones were then cut with *Pac*I and transfected into Ad-293 cells using Viral-Pack Transfection Kit from Stratagene. Viral titers were determined by End-Point Dilution Assay. MOI of 100 was used in the infections.

For lentiviral vectors, Drp1 S637A mutant was amplified from AAV-DIO-Drp1 (S637A)-EYF mutation vector (Gift from Dr. Lobo at University of Mariland), respectively, using primers: (5′ AATGGTACCATGGAGGCGCTAATTCCT3′, and 5′AAACTCGAGTCACCAAAGATGAGTCTCCCGGA3′). Lenti-GFP or lenti-Drp1637A was transduced into control or PGAM5$^{-/-}$ APRE19 cells at M.O.I. of 10. Immunostaining and western blot were performed at 48 h and 1 week after transduction respectively. ATP level was measured at one week (short-term) and eight weeks (long-term). SA-β-Gal assay was carried out at eight weeks after transduction.

**Cell culture, treatments and assays**. Wild-type and *PGAM5*$^{-/-}$ ARPE-19 cells were cultured in Dulbecco's modified Eagle's medium (DMEM)/Ham's F12 50/50 mix medium supplemented with 10% FBS (Sigma-Aldrich), 100 µg/ml penicillin and 100 µg/ml streptomycin (Gibco). Fresh medium was changed every 3 days. Cells were harvested in one week after reaching confluence for short-term culture. For long-term culture for cellular senescence, cells were cultured for 8–12 weeks after reaching 100% confluence. Primary human RPE (hRPE) cells (Lonza) were cultured in low glucose DMEM/Ham's F12 50/50 mix medium supplemented with 10% FBS, 100 µg/ml penicillin and 100 µg/ml streptomycin. HUVEC cells (ATCC) were cultured in EC growth medium EGM-2 (Lonza). For hRPE and HUVEC cell culture, 0.1% Gelatin was used to coat cell culture plates.

For cell treatment, Cycloheximide (10 mg/ml in DMSO, −20 °C), Oligomycin A (10 mM in DMSO, −20 °C), Sodium Iodate (NaIO$_3$, 10 mg/ml), and N-acetyl-L-cysteine (NAC) (3 mM in water, 4 °C) were from Sigma-Aldrich (St. Louis, MO, USA). For in vitro NaIO$_3$ treatment, 5–80 mM NaIO$_3$ was added into ARPE-19 cells for overnight, and cell viability was evaluated by Propidium Iodide (PI) staining and neutral red assays. For in vitro CCCP treatment, 20 µM CCCP was added in to medium for indicated duration. Samples were harvested for western blot or other subsequent assays. For PI staining, 10 µg/ml of PI was added to the fresh medium and incubated for 10 min before imaging at 550 nm. Neutral red assay method was adapted from[66]. Briefly, phenol red-free DMEM (Gibco) containing 0.4 mg/ml neutral red (Sigma-Aldrich) was incubated with cells for 2 h in 37 °C incubator. After incubation, cells were washed three times by PBS buffer and a solubilization solution (50% ethanol/1% acetic acid) was added. Concentration of neutral red was measured by absorbance at 540 nm. For JC-1 assay, 2 µM of JC-1 dye was added into cell culture medium. After 15 min of incubation at 37 °C, medium was removed, and fresh medium was added. Then, cell images were captured under the microscopy immediately by 488 nm and 555 nm and images were analyzed by NIS-Elements software. Cellular diameter was measured using NIS-Elements software.

SA-βGal staining in cultured cells was performed similarly with previous study[67]. Briefly, after fixation with 4% PFA for 10 min, cells were washed three times by PBS buffer (pH 7.4) and then incubated with freshly prepared staining solution (5 mM Potassium Ferricyanide Crystalline (Sigma-Aldrich), 5 mM Potassium Ferricyanide Trihydrate (Sigma-Aldrich), 2 mM Magnesium Chloride (Sigma-Aldrich), 1 mg/ml X-gal (Sigma-Aldrich) in PBS buffer (pH 6.0)) for 12–24 h. After removing staining solution and washing cells with PBS buffer (pH 7.4), bright field microscopy was used to take the images.

Cellular ATP level was measured as described by CellTiter-Glo® 2.0 Cell Viability Assay Kit (Omega)[68]. Briefly, for 96-well plates, 100 µl of Glo2.0 buffer was added into 100 µl culture medium. After 10 min, luminescence was read using SpectraMax iD3 (Molecular Devices) plate reader.

Reactive oxygen species (ROS) was measured using DCFDA-Cellular ROS Assay Kit following the manufacture's protocol (Abcam, ab113851). Cells were counted and seeded to 96-well plates before growing to 100% confluence. After changing medium for 2 days, cells were washed by washing buffer and stained by DCFDA for 45 min at 37 °C in the dark. Plates were read on a SpectraMax iD3 (Molecular Devices) plate reader at Excitation length 485 nm and emission length of 535 nm.

Mitotimer assay was used to monitor mitochondrial turnover as described[69]. pTRE-Tight-MitoTimer plasmid (addgene#50547) was transfected into ARPE-19 cells using lipofectamine 3000. After 24 h, 0.5 µg/ml doxycycline was added to induce Mitotimer expression for 4 h before changing into fresh medium. Cell was

fixed after 12 h, images were captured at 488 nm and 555 nm using Nikon A1 confocal microscopy and were analyzed as described[69].

**Western blot analyses**. For western blot, confluent cells were lysed in lysis buffer (CellLytic™ Tissue lysis Reagent (Sigma-Aldrich) and 2× Laemmli Sample Buffer (Bio-rad) at 1:1 ratio, protease inhibitor cocktails, phosphatase inhibitor cocktails). Cell lysate was sonicated using ultrasonic cleaner (KENDAL) for 10 min before boiling for western blot. After electrophoresis and transferring, PVDF membranes were probed by primary and secondary antibodies and detected by Odyssey Infrared Imaging System (LI-COR). For in vivo samples, RPE/choroid cells were harvested from mice eye cups and treated as above.

Primary antibodies used in this study include: anti-Tom20 (1:1000, BD, 612278), anti- P16Ink4a (1:1000, BD Biosciences, 550834), anti-cytochrome *C* (1:1000, Abcam, ab110325), anti-phospho-IRF-3 (Ser396) (1:1000, Cell Signaling Technologist, 4947) anti-CYPD (1:1000. Abcam, ab110324), anti-α-tubulin (Millipore, 05-829), anti-β-actin (1:2000, Santa Cruz, sc-47778), anti-PGAM5 (1:1000, Santa Cruz, A-3), anti-p-Drp1637 (1:1000, Abcam, ab193216), anti-pS6 (1:1000 for WB, 1:400 for IF, Cell Signaling, 2211), anti-S6 (1:1000, Cell Signaling, 2317), anti-phosphor-AMPK(172) (1:1000, Cell Signaling, 2535), anti-total-AMPK (1:1000. Cell Signaling, 2532), anti-phosphor-TSC2(1387) (1:1000, Cell Signaling, 5584T), anti-Lamin B1 (1:1000, Santa Cruz, sc-377000), anti-MMP3 (1:1000, Chemicon International, MAB1339), anti-macroH2A (1:1000, Abcam, ab37264) anti-phosphor-4EBP1 (37/46) (1:1000, Cell Signaling, 28555), anti-Axin1 (1:1000, Cell Signaling, 2087S), anti-Nrf2 (1:1000, Santa Cruz,sc-365949) anti-Drp1 total (1:1000, Abcam, ab56788), anti-gamma H2A.X (phosphor S139) (1:400, Abcam, ab81299), anti-Drp1 total (1:1000, Santa Cruz, sc-271583), phoillodin-488 (1:400, Cell Signaling, 8878), anti-Zo1 (1:400, Invitrogen, MA3-39100-A647), anti-IBA1 (1:400, FUJIFILM, 019-19741), anti- P16Ink4a (1:400, Abcam, ab54210). Secondary antibodies for western blot were from LI-COR Biosciences (Lincoln, NE, USA): goat anti-Mouse 800 secondary antibody (1:5000, Licor, 926-32210), goat anti-Rabbit 680 secondary antibody (1:5000, Licor, 926-68071).

**Histology and immunostaining**. For immunostaining in cells, after fixation, permeabilization and blocking, cells were incubated with primary antibodies overnight and followed by corresponding secondary antibodies for 2 h. Nuclei were outlined by DAPI (Sigma-Aldrich) before samples were mounted by mounting medium (Electron Microscopy Service, Hatfield, PA, USA).

To generate tissue sections for immunostaining, enucleated eye globes were cryo-preserved or fixed in Davidson's solution or 24 h. For cryosection, eyeball samples were embedded into O.C.T embedding medium (Tissue-Tek) and then transferred to −20 °C for sectioning. For parafilm section, samples were processed for dehydration and embedding. When cutting, 10 µm thickness was set and about 100 sections were collected for each eye, and only sections close to or crossing optic nerve were used for histology and staining. Sections were stained with hematoxylin and eosin (H&E) to visualize RPE morphology. For immunostaining on sections, antigen retrieval was performed by using citrate buffer (10 mM Citric Acid, 0.05% Tween 20). After blocking, sections were incubated with primary antibodies overnight, and with secondary antibodies for 2 h after washing. Primary antibodies include: anti-Tom20 (BD, 612278), phospho-anti-γH2AX (Abcam, ab81299), pholloidin-488 (Cell Signaling, 8878), ZO-1 antibody (Invitrogen, RE238007), anti-P16$^{Ink4a}$ (BD Biosciences, 550834), anti-pS6 (Cell Signaling, 2211), anti-Iba1 (Wako Pure Chemical Industries). Secondary antibodies for immunofluorescence were from Invitrogen: goat anti-mouse 549, goat anti-mouse 488, goat anti-rabbit 549, goat anti-rabbit 488. DAPI was used to counterstain the nuclei.

For mice flatmount staining, whole mount of RPE/choroid/sclera tissues were dissected from mice and fixed by 4% paraformaldehyde (PFA) for 2 h or cold methanol (−20 °C) for 30 min and permeabilized by 0.5% Triton X-100 at room temperature for 2 h. After blocking in 3% (v/v) fetal bovine serum PBS buffer, flatmounts were incubated with primary antibodies for overnight, followed by corresponding secondary antibodies for 2 h. Before mounting the flatmounts with mounting medium (Electron Microscopy Service, Hatfield, PA, USA), nuclei were counterstained by DAPI (Sigma-Aldrich).

All images were captured using Nikon microscope or Nikon A1 confocal microscopy system. Data were analyzed by Nikon Workstation System and ImageJ.

**Quantitative reverse transcription PCR (qRT-PCR)**. For cultured cells, total RNA was extracted by Trizol reagent (Invitrogen). After quantification and DNAase treatment, cDNA was made using an iScript™ cDNA synthesis kit (Bio-Rad). qRT- PCR was carried out using SYBR™ Green PCR Master Mix (Thermo-Fisher Scientific). β-actin or GAPDH was used as normalization control for PCR. For tissue, RPE/choroid complex was detected from eye cups. After brief centrifugation, cells were suspended and lysed in Trizol reagent before processing for qRT-PCR (see Supplementary Table 1 for PCR primers).

**Co-immunoprecipitation assay**. For co-immunoprecipitation assay, 1 ml lysis buffer (20 mM Tris-HCl pH 8, 137 mM NaCl, 1% IGEPAL CA-630, 0.1% Triton X-100, 2 mM EDTA) supplemented by protease inhibitor (Pierce) was added into 100 mm dish to lyse the ARPE-19 cells on ice. About 10$^7$ cells (from 2 dishes) were pooled together. 100 µl of the supernatant of cell lysate was kept as input, and the

rest were incubated with 3 µg Axin1 antibody (Cell Signaling Technologist) or control rabbit IgG (Santa Cruz sc-66931) at 4 °C for overnight. 30 µl of protein A and G conjugated magnetic beads (Bio-rad 1614833) were washed by lysis buffer and added into antibody–cell lysate mixture for 4 h incubation at 4 °C. Magnetic rack was used to precipitate the magnetic beads, and beads were subsequently rinsed by pre-chilled lysis buffer for three times. 1× SDS sample loading buffer (Bio-rad) was used to elute the proteins. Western blot was used for the downstream assay.

**Mitochondrial DNA copy number**. mtDNA copy number was counted as described[70]. Briefly, 48 h after treatments and medium change, ARPE-19 cells were lysed in 25 mM NaOH, 0.2 mM EDTA, pH 12.3 for 5 min at room temperature. 20 µl of lysate were heated at 95 °C for 15 min and neutralized by 20 µl 40 mM Tris-HCl (pH 4.9). After dilution at 1:64, qPCR was performed to measure mtDNA copy. For the system: TaqMan universal PCR mastermix (Applied Biosystems), and mtND2 probe mix (900 nM primers, 250 nM probe) for mitochondrial DNA, and Alu probe mix (1000 nM primers, 100 nM probe) for genomic DNA. Sequence for primers and probes: mtND2-F: 5′-tgttggttatacccttcccgtacta-3′; mtND2-R: 5′-CCTGCAAAGATG GTAGAGTAGATGA-3′; probe-6FAM: CCCTGGCCCAACCC; Alu-F: 5′-CTTG CAGTGAGCCGAGATT-3′; Alu-R: 5′-GAGACGGAGTCTCGCTCTGTC-3′; probe-HEX$^{TM}$: ACTGCAGTCCGCAGTCCGGCCT.

**Cell size measurement**. For in vitro cell size quantification, cells were digested with trypsin and suspended, and images were captured by microscopy. Cell size was also quantified by flow cytometry using up to $10^5$ cells (Sony SH800S Cell Sorter). Fordware Scatter (FSC) parameter along the $X$ axis of flow cytometry result represents the relative size of cells.

For in vivo RPE cell size quantification, RPE flatmounts were stained by phoillodin or ZO-1 antibody to outline the boundaries of cells, and cell numbers were counted. Average cellular area was calculated based on the number of cells in the area.

**Statistics**. Graphpad Prism 6 was used to perform statistical analyses. For most experiments, two-tailed unpaired $t$-tests and Tukey's multiple comparisons two-way ANOVA were used to determine statistically significant differences between groups. All experiments were performed at least three times and data were presented as means ± standard deviation (s.d.) unless otherwise specified. Exact $P$-values were labeled in figure legends. The details of each experiment could be referred to the corresponding figure legends.

**Reporting summary**. Further information on research design is available in the Nature Research Reporting Summary linked to this article.

## Data availability
The source data underlying Figs. 1–6, 8, 9 and Supplementary Figs. 1–6 are provided as a Source Data file. Other figures or data supporting the results of this study are available from the corresponding author upon any reasonable request.

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

## Acknowledgements
We thank Dr. Yiping Chen and Dr. Frank Jones for encouragement and for sharing equipment, Dr. Mary Kay Lobo from University of Maryland for providing Drp1S637A plasmids, and Dr. Wang Wang from University of Washington for providing Drp1-K38A plasmids. S.W. was supported by a Startup fund from Tulane University, NIH Grants EY021862 and EY026069.

## Author contributions
B.Y. and S.W. designed the study. B.Y., J.M. and J.L. executed the experiments. B.Y. and J.M. maintained the mice line. J.L. performed the bioinformatics analysis. D.W. and Z.W. supplied *Pgam5⁻/⁻* mice and participated in study design and result discussion. S.W. and B.Y. wrote the paper with input from all authors.

## Competing interests
The authors declare no competing interests.
