## [Peer Review File · Nature Communications]

Reviewers' comments:

Reviewer #1 (Remarks to the Author):

This research focused on the role of mitochondrial phosphatase PGAM5 in the regulation of mitochondrial dynamics and cellular senescence. PGAM5 was shown to be a regulator of mitochondrial fission by dephosphorylating DRP1, thus lead to elevated ROS generation, mTOR activation and accelerated cell senescence.

However, I have some concerns about the study.

Major points:

1. The main concern is about the significance of the study. The authors do discovered that PGAM5 deletion can cause mitochondrial hyperfusion and cell senescence and established the connection between PGAM5 deletion and mitochondria dynamics. However, these data are not enough for proving the causal relationship between PGAM5 and cell senescence or AMD. As many genetic manipulations, particular those with the deletion of a functional mitochondrial protein, can accelerate senescence. I think the role of PGAM5 in senescence demonstrated by this work is reasonable but not intriguing. And most problematically it is not solid. I mean it needs to be confirmed by other sets of evidence in principle, but not only those based on genetic manipulation. For example, the authors may present additional data to show whether PGAM5 expression or activity altered during cell senescence or during the development of AMD.

2. The logic route of this manuscript is not concentrated. For instance, it mentioned about the matter of mitophagy in abstract as a result, but no any evidence was presented in the manuscript. Given authors talked about mitophagy also in introduction and discussion, I guess this is a point they are interested in but not clear yet? The other panels belonging to so called mechanistical analysis are also scattered, the causal association among AMPK, mTOR, ROS and others did not form a completed story.

3. The authors claimed that PGAM5 modulated cell senescence by regulating mitochondrial dynamics. However, it seems no rescue experiment is conducted. Although overexpression of Drp1 K38A mimics PGAM5 deletion, it is not sufficient to link PGAM5 deletion and cell senescence together, especially verify the contribution of mitochondrial fission inhibition. Authors may explore whether the pro-senescence phenotype can be rescued when mitochondria fission is evoked in PGAM5 KO mice/cells.

Minor points:

1. The quality of some immunoblots is poor and needs to be improved, such as othse in Figure1C, Figure2A, Figure4A, Figure4C, Figure7H.

2. The label is inconsistent between figures (e.g. actin/b-actin).

3. β -gal staining in Figure 2B requires statistical analysis.

4. In Figure 3H, the magnification in WT and PGAM5-KO is not consistent.

Reviewer #2 (Remarks to the Author):

Manuscript #: NCOMMS-19-036627

Title: Mitochondrial phosphastase PGAM5 modulates cellular senescence by regulating mitochondrial

dynamics

Authors: Bo Yu, Jing Ma, Dazhi Wang, Zhigao Wang, Shusheng Wang

General Comments:

Strengths - The authors present convincing data showing that PGAM5 deficiency induces senescence. There are many well-controlled experiments that are presented clearly in the graphs and immunohistochemical images. The data represent studies of 3 different cell types in vitro and in vivo work with knockdown mice. The authors examine short term studies and longer time periods in their work. It is a very solid body of work. The findings in this paper will be of interest to researchers in the mitochondria field because of their importance in age-related diseases. Understanding the mechanisms for maintaining mitochondria homeostasis is critical for future therapies.

Weakness - The Results section is written as a Discussion. It has references, interpretations and summaries. Normally the Results section presents the data but does not give background and interpretation for the results. With the constant interjection of background information and interpretation, it makes it harder to follow just the data itself. The combining of Results with Discussion may be very acceptable to the journal's format. It is reasonable to have some background as to why an experiment is designed a certain way. However, this reviewer thinks some of the interpretation could be moved to the Discussion. Since you do not want to make it a much longer paper, perhaps the authors could put subheadings for each section and summarize the findings in the Discussion and leave the Results for the data.

Specific Comments:

1. Page 4, line 9. "senescence, Inhibition". Should not be capitalized, should be "senescence, inhibition"
2. Page 8, paragraph 1, line second to last sentence. Please standardize the font.
3. Page 9, Figure 2B-2C – It is difficult to state that the volume is bigger based upon the phase contrast image alone. Were other methods used to quantify the volume differences?
4. Page 9, Figure 2D. With the KO panels, what is the lower molecular weight band in the MMP3 image? Is this the active form? Are there 2 isoforms of the preform? Please clarify.
5. Page 9, Move the "(Fig. 2F)" to after the sentence on IL6 and MMP3. It should not be after the TNF α sentence.
6. Page 9, Figures 1 and 2. The authors presents convincing data showing that PGAM5 deficiency induces senescence.
7. Page 11, paragraph 1. (a) Page 10, line 1. refers to Fig. 3E but the figure shows mtDNA copy numbers. Please correct the Figs in text so they match the actual figures. (b) Page 11, line 2, refers to Fig. 3F. But the figure shows CYPD, PGAM5, b-Tubulin, not mtDNA copy numbers. Please correct the figures in the text so they match the actual Figures. (c) Also check the Fig. 3G image to see it matches the text.
8. Page 12, paragraph 3, lines 1-2. The ATP increase was only at 1 week but declined by 8 weeks. At what time point was the AMPK-mTOR studies conducted?
9. The HRPE cells (Supp. Fig 3A) has multiple bands for the S6 western blot. The HUVEC (Supp. Fig. 3B) shows a single band for S6. What do the authors conclude is the difference in the patterns?

10. Authors state that the Figure 6C-4 of the middle region shows minimal RPE damage but the right-hand side of the images shows disrupted RPE cells. I agree that the Periphery Fig 6, C-6 looks good and can be considered minimal damage but the C-4 image should be acknowledged as some RPE damage.

11. Figures 7D and 7E. Why examine SOD1 instead of the mitochondria-associated SOD2?

13. It is hard to understand how the 18-month PGAM5 KO values can be significantly different with the error bars presented in Figure. 8F. Please re-check the values.

14. Figure Legends 3E - 3G are not representing the correct images. Please re-organize.

15. Fig. 6. legend needs to be placed on a new line. It is running onto the last sentence of Fig 5.

Reviewer #3 (Remarks to the Author):

Yu et al. present new findings related to the role of the mitochondrial Phosphatase PGAM5 in modulating mitochondrial dynamics and cellular senescence during aging.

This is a very interesting and active research area. Numerous recent studies in yeast, worms and flies have examined the role of mitochondrial dynamics in aging, mitophagy and lifespan determination. However, there is no clear (or simple) take home message from these studies across species. That said, two recent studies in *Drosophila* do support the idea that promoting mitochondrial fission can facilitate mitophagy and slow aging in *Drosophila*:

Rana et al Nat. Comm. 2017

Aparicio et. al. Cell Reports 2019

The authors present new finding in mice which support the following model:

Loss of PGAM5 leads to loss of mito fission and therefore impaired mitophagy. This, in turn, leads to mTOR activation and cell senescence.

The data appear to be well-controlled and carefully carried out. The data adds to our understanding of the interplay between mito dynamics and mitophagy during aging.

I do, however, have some conceptual concerns regarding novelty and suggestions to improve the paper:

Novelty/conceptual concerns:

It was previously shown that PGAM5 is involved in mitophagy. Is it really surprising/novel that defective mitophagy (in PGAM5 $-/-$ mice) would show early-onset senescence?

It is always difficult to exclude the possibility that these mice show a novel pathology, as opposed to 'accelerated aging'.

The novelty/impact of the study would be greatly improved if there was an intervention to restore mitophagy during aging and, thereby, delay senescence.

Minor concerns regarding intro/discussion:

In the intro, the authors state that "Maintenance of the fused mitochondrial network in *C. elegans* is necessary for longevity, and absence of mitochondrial fusion in the *Deltamgm1* mutant leads to a striking reduction of both replicative and chronological lifespan in *C.elegans* (11, 12).

Ref 11 is a yeast study. Not a *C. elegans* study.

Ref 14, it may be worth pointing out "Promoting Drp1-mediated mitochondrial fission in midlife FACILITATES MITOPHAGY and prolongs healthy lifespan of *Drosophila melanogaster* (14)". This appears to be CONSISTENT with the authors' data presented herein.

Perhaps in the discussion, the authors may consider trying to discuss how their findings fit with the previous work in worms and flies.

Itemized response to the review comments:

I would like to thank the reviewers and editors for their positive comments and insightful suggestions. Further experiments and editorial changes have been performed to address all the questions raised. I hope you will find the paper is significantly improved with the revision. The changes are highlighted in red in the manuscript. Itemized response to the reviewer comments are detailed as below.

Best regards,

Shusheng Wang, PhD/MBA

Reviewers' comments:

Reviewer #1 (Remarks to the Author):

This research focused on the role of mitochondrial phosphatase PGAM5 in the regulation of mitochondrial dynamics and cellular senescence. PGAM5 was shown to be a regulator of mitochondrial fission by dephosphorylating DRP1, thus lead to elevated ROS generation, mTOR activation and accelerated cell senescence. However, I have some concerns about the study.

Major points:

1. The main concern is about the significance of the study. The authors do discovered that PGAM5 deletion can cause mitochondrial hyperfusion and cell senescence and established the connection between PGAM5 deletion and mitochondria dynamics. However, these data are not enough for proving the causal relationship between PGAM5 and cell senescence or AMD. As many genetic manipulations, particular those with the deletion of a functional mitochondrial protein, can accelerate senescence. I think the role of PGAM5 in senescence demonstrated by this work is reasonable but not intriguing. And most problematically it is not solid. I mean it needs to be confirmed by other sets of evidence in principle, but not only those based on genetic manipulation. For example, the authors may present additional data to show whether PGAM5 expression or activity altered during cell senescence or during the development of AMD.

Thank you for your insightful comments on our study and the agreement about our genetic

model. We respectfully disagree that our work is not solid. Our data did provide genetic evidence that PGAM5 regulates cell senescence through modulating mitochondrial dynamics. We have included additional evidence showing that overexpression of Drp1 S637A phosphorylation mutant is able to rescue the mitochondrial and senescence phenotypes in PGAM5 knockout cells (See New Figure 6), further supporting our central hypothesis. The significance of the paper reflects the fact that we found defective mitochondrial dynamics could lead to senescence in mice, and this could be regulated by PGAM5. Currently, the previously published data on this subject have been conflicting, or only providing evidence in fly or worms.

Establishing PGAM5 as critical mediator of mitochondrial dynamics and senescence does not suggest that PGAM5 itself is regulated during senescence or AMD. Although this project may have implications in AMD, our study did not aim to establish a direct role of PGAM5 in AMD. Nevertheless, mutations in PGAM5 could link to senescence or AMD, which will be studied in the future. Our long-time goal is to establish the causal relationship between mitochondrial dynamics and AMD, which is not within the scope of the current study.

We have attempted to detect PGAM5 expression level in existing datasets GSE29801 and GSE50195. There is no significant difference between the young (<60 years) and the elder (>60 years)(Fig R1). There is neither significant difference between AMD patients and healthy people, as shown below (Fig R2, R3). We would acknowledge that these are non-exhaustive transcriptional level studies, not necessarily reflecting PGAM5 activity or cleavage. Many studies have showed that mitochondrial mass accumulation and compromised mitochondrial dynamics during cell ageing (1), which was consistent with what we have observed.

Fig R1

Fig R2

Fig R3

2. The logic route of this manuscript is not concentrated. For instance, it mentioned about the matter of mitophagy in abstract as a result, but no any evidence was presented in the manuscript. Given authors talked about mitophagy also in introduction and discussion, I guess this is a point they are interested in but not clear yet? The other panels belonging to

so called mechanistical analysis are also scattered, the causal association among AMPK, mTOR, ROS and others did not form a completed story.

The complex relationship between PGAM5 and mitophagy has been demonstrated by several other groups in past years. For example, Chen G *et.al* confirmed PGAM5 regulated mitophagy through dephosphorylating Ser-13 at FUNDC1(2); Park *et.al* proved PGAM5 regulated PINK1/Parkin-mediated mitophagy via Drp1(3). Both of them focused on the mitophagy-directly-associated receptor/proteins. We focused on PGAM5 function in mitochondrial fission and its consequence in senescence. Others have shown that mitochondrial fission is required for mitophagy (4). Drp1 is the master regulator for mitochondrial fission (5). Our data provided *in vivo* evidence that PGAM5 is required for mitochondrial fission and prevention of senescence, and Drp1 plays a critical role in mediating PGAM5 function. Mechanistically, we found that cleaved PGAM5 recruits Drp1 for its dephosphorylation (Fig. 3C-D, SFig. 2C), representing a mechanism to regulate mitochondrial fission and mitophagy under stress condition.

Our mechanistical analysis may appear to be scattered. However, it was centered on consequences resulted from the mitochondrial hyperfusion by PGAM5 deletion. We found elevated ATP but not ROS is required for the stimulated mTOR pathway. This established a mechanistic link between mitochondrial hyperfusion and mTOR pathway that is known to be critical for senescence. In addition, we found ROS increase and IRF3 pathway activation, both of which have been shown to drive cell senescence by other groups. Therefore, we discovered a center hub that can regulate different known drivers for senescence.

3.The authors claimed that PGAM5 modulated cell senescence by regulating mitochondrial dynamics. However, it seems no rescue experiment is conducted. Although overexpression of Drp1 K38A mimics PGAM5 deletion, it is not sufficient to link PGAM5 deletion and cell senescence together, especially verify the contribution of mitochondrial fission inhibition. Authors may explore whether the pro-senescence phenotype can be rescued when mitochondria fission is evoked in PGAM5 KO mice/cells.

We thank the reviewer's suggestion. In the revised manuscript, we have included the rescue experiments showing overexpression of Drp1-S637A rescued PGAM5 knockout phenotype in mitochondrial hyperfusion and senescence (see new Fig. 6).

Minor points:

1. The quality of some immunoblots is poor and needs to be improved, such as othse in Figure1C, Figure2A, Figure4A, Figure4C, Figure7H.

We thank reviewer's comments. We have replaced the low-quality figures in our revised manuscript.

2. The label is inconsistent between figures (e.g. actin/b-actin).

Thank you. We corrected the labels in revised manuscript.

3. β -gal staining in Figure 2B requires statistical analysis.

Thank you. We added the statistical analysis.

4. In Figure 3H, the magnification in WT and PGAM5-KO is not consistent.

Thank you for your comments. We checked the magnification in Fig. 3H and they are consistent.

Reviewer #2 (Remarks to the Author):

Manuscript #: NCOMMS-19-0366627

General Comments:

Strengths - The authors present convincing data showing that PGAM5 deficiency induces senescence. There are many well-controlled experiments that are presented clearly in the graphs and immunohistochemical images. The data represent studies of 3 different cell types in vitro and in vivo work with knockdown mice. The authors examine short term studies and longer time periods in their work. It is a very solid body of work. The findings in this paper will be of interest to researchers in the mitochondria field because of their importance in age-related diseases. Understanding the mechanisms for maintaining mitochondria homeostasis is critical for future therapies.

Weakness - The Results section is written as a Discussion. It has references, interpretations and summaries. Normally the Results section presents the data but does not give background and interpretation for the results. With the constant interjection of background information and interpretation, it makes it harder to follow just the data itself. The combining of Results with Discussion may be very acceptable to the journal's format. It is reasonable to have some background as to why an experiment is designed a certain way. However, this reviewer thinks some of the interpretation could be moved to the Discussion. Since you do not want to make it a much longer paper, perhaps the authors could put subheadings for each section and summarize the findings in the Discussion and leave the Results for the data.

Thank you for pointing out the strengths and weaknesses of the paper. We have reduced the discussion in the results portion as suggested by the review. However, we feel some introduction related to the results section needs to be kept in the result section, since there

are specifically related to the results but not the whole paper. Including interpretations in the result section may help readers better understand the science behind the experiments.

Specific Comments:

1. Page 4, line 9. "senescence, Inhibition". Should not be capitalized, should be "senescence, inhibition"

Thank you. We corrected it in revised manuscript.

2. Page 8, paragraph 1, line second to last sentence. Please standardize the font.

Thank you. We corrected it in the revised manuscript.

3. Page 9, Figure 2B-2C – It is difficult to state that the volume is bigger based upon the phase contrast image alone. Were other methods used to quantify the volume differences?

Thank you for your comments. We quantified the cellular volume by two methods: one is the flow cytometer as shown in Figure 2C, which is based on a widely accepted concept that the FSC reflect the cellular size; we also measured the cell diameter based on the phase contrast images from the isolated cells using software (in New Supplementary Fig1C)

4. Page 9, Figure 2D. With the KO panels, what is the lower molecular weight band in the MMP3 image? Is this the active form? Are there 2 isoforms of the preform? Please clarify.

Thank you for your comments. We checked the NCBI database and literatures. We didn't find MMP3 has any isoforms or active form. We suspected the lower band was due to Western blot transfer, samples treatment (boiling) or storage (-80°C). To clarify this, we redid this experiment and updated the MMP3 band in revised version.

5. Page 9, Move the "(Fig. 2F)" to after the sentence on IL6 and MMP3. It should not be after the TNF α sentence.

Thank you. We adjusted it.

6. Page 9, Figures 1 and 2. The authors presents convincing data showing that PGAM5 deficiency induces senescence.

Thank you for your suggestion. We refined our description according to your suggestion.

7. Page 11, paragraph 1. (a) Page 10, line 1. refers to Fig. 3E but the figure shows mtDNA copy numbers. Please correct the Figs in text so they match the actual figures. (b) Page 11,

line 2, refers to Fig. 3F. But the figure shows CYPD, PGAM5, b-Tubulin, not mtDNA copy numbers. Please correct the Figures in the text so they match the actual Figures. (c) Also check the Fig. 3G image to see it matches the text.

Sorry for the inconvenience due to the unintentional errors. We corrected them in the revised version.

8. Page 12, paragraph 3, lines 1-2. The ATP increase was only at 1 week but declined by 8 weeks. At what time point was the AMPK-mTOR studies conducted?

AMPK-mTOR pathway was studied at 1 week. We included this information in revised manuscript.

9. The HRPE cells (Supp. Fig 3A) has multiple bands for the S6 western blot. The HUVEC (Supp. Fig. 3B) shows a single band for S6. What do the authors conclude is the difference in the patterns?

Due to different gels, or some unknown reasons in Western blot transfer, sample treatment, the different patterns occurred sometimes even we used the same antibodies. To clarify the misunderstanding, we carried out this experiment using HRPE cells again, and the new images were included in Supp.Fig3A.

10. Authors state that the Figure 6C-4 of the middle region shows minimal RPE damage but the right-hand side of the images shows disrupted RPE cells. I agree that the Periphery Fig 6, C-6 looks good and can be considered minimal damage but the C-4 image should acknowledged as some RPE damage.

Thank you. We changed the description in revised text.

11. Figures 7D and 7E. Why examine SOD1 instead of the mitochondria-associated SOD2?

In our project, PGAM5 deletion leads to ROS elevation and oxidant-stress resistance in RPE cells. We hypothesized that the oxidative stress resistance is due to a global feedback activation of antioxidative genes, but not just the mitochondrial associated genes. Upregulation of SOD1, NRF2, and FOXO4 expression verified our initial hypothesis. Here, we analyzed SOD2 gene expression and the result as below (Fig. R4, R5). We can see, like SOD1, NRF2 and FOXO4, SOD2 also increased about 4 folds in the early passage (1 week). While, different from other genes decreasing at late passage (8 weeks), SOD2 still maintain about 2 folds of expression comparing with control group, although the relative fold was decreased from 4 to 2. This suggested the antioxidant response gene expression may not be synchronous after PGAM5 deletion.

Fig R4

Fig R5

13. It is hard to understand how the 18-month PGAM5 KO values can be significantly different with the error bars presented in Figure. 8F. Please re-check the values.

We understand your concern. We performed the statistical assay by using GraphPad software and it did show significant difference. In the revised manuscript, we dissected more mice and pooled the data together. The new quantification data was updated.

14. Figure Legends 3E - 3G are not representing the correct images. Please re-organize.

Sorry for the inconvenience and we re-organized them in our manuscript.

15. Fig. 6. legend needs to be placed on a new line. It is running onto the last sentence of Fig 5.

Sorry. We corrected it.

Reviewer #3 (Remarks to the Author):

Yu et al. present new findings related to the role of the mitochondrial Phosphatase PGAM5 in modulating mitochondrial dynamics and cellular senescence during aging.

This is a very interesting and active research area. Numerous recent studies in yeast, worms and flies have examined the role of mitochondrial dynamics in aging, mitophagy and lifespan determination. However, there is no clear (or simple) take home message from these studies across species. That said, two recent studies in Drosophila do support the idea that promoting mitochondrial fission can facilitate mitophagy and slow aging in Drosophila:

Rana et al Nat. Comm. 2017

Aparicio et. al. Cell Reports 2019

The authors present new finding in mice which support the following model:

Loss of PGAM5 leads to loss of mito fission and therefore impaired mitophagy. This, in turn, leads to mTOR activation and cell senescence.

The data appear to be well-controlled and carefully carried out. The data adds to our understanding of the interplay between mito dynamics and mitophagy during aging.

Thank you for your positive comments.

I do, however, have some conceptual concerns regarding novelty and suggestions to improve the paper:

Novelty/conceptual concerns:

It was previously shown that PGAM5 is involved in mitophagy. Is it really surprising/novel that defective mitophagy (in PGAM5 $-/-$ mice) would show early-onset senescence?

We understand the reviewer's concern. Indeed, the relationship between PGAM5 and mitophagy was investigated before (2, 3), and that's why we didn't take the process as main objective in our project, but focused on its consequence. Linking defective mitophagy to early-onset senescence is not totally new. However, as stated by the review, there have been controversy in the field regarding the role of mitochondrial fission and fusion in senescence. Most of the previous models were based on studies in fly or worm. Our model is based on mouse model, which is novel to some extent, and could have better implications in humans. In addition, we found that mitochondrial hyperfusion leads to mTOR signaling activation, which has novelty and could explain the phenotype of accelerated senescence caused by mitochondrial hyperfusion.

It is always difficult to exclude the possibility that these mice show a novel pathology, as opposed to 'accelerated aging'.

Thank you for your comments. We agreed with this comment to some extent. For *Pgam5*^{-/-} mice, accelerated senescence is one of the most interested phenotypes observed by us. In the adult mice, we did observe some other pathologies, including seizure, hunched posture, firm masses in abdomen, dermatitis on rear legs and head tilt, *et.al*. Most of them could be "age-related" phenotypes in our opinion. We have not observed other direct and obvious phenotypes besides the phenotypes mentioned above.

The novelty/impact of the study would be greatly improved if there was an intervention to restore mitophagy during aging and, thereby, delay senescence.

We appreciated your advice. We have performed the "rescue" experiments as shown in New Fig. 6.

Minor concerns regarding intro/discussion:

In the intro, the authors state that "Maintenance of the fused mitochondrial network in *C. elegans* is necessary for longevity, and absence of mitochondrial fusion in the *Deltamgm1* mutant leads to a striking reduction of both replicative and chronological lifespan in *C. elegans* (11, 12).

Ref 11 is a yeast study. Not a *C. elegans* study.

Ref 14, it may be worth pointing out "Promoting Drp1-mediated mitochondrial fission in midlife FACILITATES MITOPHAGY and prolongs healthy lifespan of *Drosophila melanogaster* (14)". This appears to be CONSISTENT with the authors' data presented herein.

Perhaps in the discussion, the authors may consider trying to discuss how their findings fit with the previous work in worms and flies.

Thank you for your suggestions. We discussed the consistency and difference between our results and previously published work extensively in our discussion in revised manuscript. Please refer to the new version.

1. Sharma A, Smith HJ, Yao P, and Mair WB. Causal roles of mitochondrial dynamics in longevity and healthy aging. *EMBO Rep.* 2019;20(12):e48395.
2. Chen G, Han Z, Feng D, Chen Y, Chen L, Wu H, et al. A Regulatory Signaling Loop Comprising the PGAM5 Phosphatase and CK2 Controls Receptor-Mediated Mitophagy. *Molecular Cell.* 2014;54(3):362-77.
3. Park YS, Choi SE, and Koh HC. PGAM5 regulates PINK1/Parkin-mediated mitophagy via DRP1 in CCCP-induced mitochondrial dysfunction. *Toxicology Letters.* 2018;284:120-8.
4. Twig G, Elorza A, Molina AJA, Mohamed H, Wikstrom JD, Walzer G, et al. Fission and selective fusion govern mitochondrial segregation and elimination by autophagy. *The EMBO Journal.* 2008;27(2):433-46.
5. Fonseca TB, Sánchez-Guerrero Á, Milosevic I, and Raimundo N. Mitochondrial fission requires DRP1 but not dynamins. *Nature.* 2019;570(7761):E34-E42.

REVIEWERS' COMMENTS:

Reviewer #1 (Remarks to the Author):

This research demonstrate their findings for the role of mitochondrial phosphatase PGAM5 in the regulation of mitochondrial dynamics and the contribution of this role to cellular senescence. There are novel evidence and intensively performed experiments. As the main results, PGAM5 was shown to be a regulator of mitochondrial fission by dephosphorylating DRP1, and its deletion led to elevated ROS generation, mTOR activation as well as accelerated cell senescence. However, I have some concerns about the study.

Major points:

1.The main concern of mine is about the significance of the study. Yes, authors discovered that PGAM5 deletion can result in mitochondrial hyperfusion and cell senescence. However, it is actually a non-spontaneous event anyway. In other words, it should be verified, at least supported by some kind of intrinsic events occurring naturally. This consideration could not be neglected when we sense the meaning of an experimental outcome, particularly with regard to cell senescence, because it is a state where the balance of cellular homeostasis disturbed and importantly, can be induced by either physiologically happened or artificially imported stimuli. In this research, regretfully, authors only presented the results based on PGAM5 deletion, without showing the physiological connection between PGAM5 and cell senescence or AMD. I think this imperfection lessened the significance of this research. The authors may present additional data to show whether intrinsic PGAM5 expression or activity altered during cell senescence or during the development of aging or AMD.

2.The authors claimed that PGAM5 modulated cell senescence by regulating mitochondrial dynamics, likely declaring the causal relationship between these two events. However, the solid evidence is missing in my opinion. For example, no relevant rescue experiment was performed. Although overexpression of Drp1 K38A mimics PGAM5 deletion in somewhat extent, it is insufficient to set the conclusion that PGAM5 deletion caused cell senescence by inhibiting mitochondrial fission, given the possibility that Drp1 may not only functioned at the downstream of PGAM5. As a suggestion, the authors can try a way to induce mitochondria fission in PGAM5 KO mice or cells, then to see whether the pro-senescence phenotype happened in KO mice can be rescued.

3.The results presented in the manuscript is scattered and do not always point to the theme. For instance, is the information about immunoactivity and IRF3/IL1B supportive for the main conclusion of the study? How do those about mTOR connected with mitochondria dynamics and senescence? They happened indeed, but what is the molecular and signaling relationship with PGAM5-mediated mitochondria dynamics and REP senescence? In a story, every piece of data should be a node standing for the theme.

Minor points:

1.The resolution of some immunoblots is poor and needs to be improved, for example, those in figure1C, figure2A, figure4A,figure4C and figure7H.

2.The labels between figures is inconsistent. e.g. actin/b-actin.

3.The β -gal staining requires statistical analysis in figure 2B.

4.In figure 3H, the magnification in WT and PGAM5-KO is not consistent.

Reviewer #2 (Remarks to the Author):

The authors have responded to my comments and the paper is acceptable for publication from this

reviewer.

Reviewer #3 (Remarks to the Author):

While my original concerns regarding the magnitude of the conceptual advance remain, the authors appear to have attempted to address reviewer concerns in a sincere fashion.

Itemized response to the review comments:

I would like to thank the reviewers and editors for their positive comments and insightful suggestions. Editorial changes have been performed to address all the questions raised. Since most of the review 1 questions have been addressed in the previous revision, we only provided the most updated information in the response letter. The changes are highlighted in red in the manuscript. Itemized response to the reviewer comments are detailed as below.

Best regards,

Shusheng Wang, PhD/MBA

REVIEWERS' COMMENTS:

Reviewer #1 (Remarks to the Author):

This research demonstrate their findings for the role of mitochondrial phosphatase PGAM5 in the regulation of mitochondrial dynamics and the contribution of this role to cellular senescence. There are novel evidence and intensively performed experiments. As the main results, PGAM5 was shown to be a regulator of mitochondrial fission by dephosphorylating DRP1, and its deletion led to elevated ROS generation, mTOR activation as well as accelerated cell senescence. However, I have some concerns about the study.

Major points:

1.The main concern of mine is about the significance of the study. Yes, authors discovered that PGAM5 deletion can result in mitochondrial hyperfusion and cell senescence. However, it is actually a non-spontaneous event anyway. In other words, it should be verified, at least supported by some kind of intrinsic events occurring naturally. This consideration could not be neglected when we sense the meaning of an experimental outcome, particularly with regard to cell senescence, because it is a state where the balance of cellular homeostasis disturbed and importantly, can be induced by either physiologically happened or artificially

imported stimuli. In this research, regretfully, authors only presented the results based on PGAM5 deletion, without showing the physiological connection between PGAM5 and cell senescence or AMD. I think this imperfection lessened the significance of this research. The authors may present additional data to show whether intrinsic PGAM5 expression or activity altered during cell senescence or during the development of aging or AMD.

Thank you, we addressed this concern in the revised version, and moved some figures from the response letters to the paper.

2.The authors claimed that PGAM5 modulated cell senescence by regulating mitochondrial dynamics, likely declaring the causal relationship between these two events. However, the solid evidence is missing in my opinion. For example, no relevant rescue experiment was performed. Although overexpression of Drp1 K38A mimics PGAM5 deletion in somewhat extent, it is insufficient to set the conclusion that PGAM5 deletion caused cell senescence by inhibiting mitochondrial fission, given the possibility that Drp1 may not only functioned at the downstream of PGAM5. As a suggestion, the authors can try a way to induce mitochondria fission in PGAM5 KO mice or cells, then to see whether the pro-senescence phenotype happened in KO mice can be rescued.

Thank you, we have included the data showing that Overexpression of S637A mutant rescues PGAM5 deletion in ATP increase, mTOR activation and senescence in the revised version.

3.The results presented in the manuscript is scattered and do not always point to the theme. For instance, is the information about immunoactivity and IRF3/IL1B supportive for the main conclusion of the study? How do those about mTOR connected with mitochondria dynamics and senescence? They happened indeed, but what is the molecular and signaling relationship with PGAM5-mediated mitochondria dynamics and REP senescence? In a story, every piece of data should be a node standing for the theme.

Thank you, we addressed this comment in the revised version. We agree that the mechanism of the IRF3 pathway activation in the PGAM5 KO mice is still unclear. DNA damage and the increase of cytoplasmic DNA may explain IRF3 activation. However, more work needs to be done to confirm. However, this support PGAM5 deletion could drive senescence through different mechanisms, and PGAM5 is the nodal point for the process.

Minor points:

1.The resolution of some immunoblots is poor and needs to be improved, for example, those in figure1C, figure2A, figure4A,figure4C and figure7H.

We thank reviewer's comments. We have replaced the low-quality figures in our revised manuscript.

2.The labels between figures is inconsistent. e.g. actin/b-actin.

Thank you, we changed it in the revised manuscript.

3.The β -gal staining requires statistical analysis in figure 2B.

Thank you. We added the statistical analysis.

4.In figure 3H, the magnification in WT and PGAM5-KO is not consistent.

Thank you for your comments. We checked the magnification in Fig. 3H and they are consistent.

Reviewer #2 (Remarks to the Author):

The authors have responded to my comments and the paper is acceptable for publication from this reviewer.

Thank you for your insightful comments during the review process.

Reviewer #3 (Remarks to the Author):

While my original concerns regarding the magnitude of the conceptual advance remain, the authors appear to have attempted to address reviewer concerns in a sincere fashion.

Thank you for your insightful comments during the review process.